# Gradient Guidance for Diffusion Models: An Optimization Perspective

**Yingqing Guo** *    **Hui Yuan** *    **Yukang Yang**    **Minshuo Chen**    **Mengdi Wang**

**Princeton University**

## Abstract

Diffusion models have demonstrated empirical successes in various applications and can be adapted to task-specific needs via guidance. This paper studies a form of gradient guidance for adapting a pre-trained diffusion model towards optimizing user-specified objectives. We establish a mathematical framework for guided diffusion to systematically study its optimization theory and algorithmic design. Our theoretical analysis spots a strong link between guided diffusion models and optimization: gradient-guided diffusion models are essentially sampling solutions to a regularized optimization problem, where the regularization is imposed by the pre-training data. As for guidance design, directly bringing in the gradient of an external objective function as guidance would *jeopardize* the structure in generated samples. We investigate a modified form of gradient guidance based on a forward prediction loss, which leverages the information in pre-trained score functions and provably preserves the latent structure. We further consider an iteratively fine-tuned version of gradient-guided diffusion where guidance and score network are both updated with newly generated samples. This process mimics a first-order optimization iteration in expectation, for which we proved $\tilde{\mathcal{O}}(1/K)$ convergence rate to the global optimum when the objective function is concave. Our code is released at https://github.com/yukang123/GGDMOptim.git.

## 1 Introduction

Diffusion models have emerged as a significant advancement in the field of generative artificial intelligence, offering state-of-the-art performance in image generation [58, 56, 22]. These models operate by gradually transforming a random noise into a structured output, utilizing the score function learned from data. One of the key advantages of diffusion models is their flexibility which allows controlled sample generation for task-specific interest, excelling diffusion models in a wide range of applications, such as content creation, sequential decision making, protein engineering [38, 1, 29, 11].

Controlling the generation of large generative models stands at the forefront of AI. Guidance and fine-tuning are two most prevalent approaches for controlling the generation of diffusion models. Unlike fine-tuning which changes the weights of pre-trained models, guidance mechanism enables a more directed and flexible control. Adding gradient-based guidance during inference was pioneered by classifier guidance [60, 22], which involves training a time-dependent classifier. Diffusion Posterior Sampling (DPS) [16] introduced a fully training-free form of gradient-based guidance, which removes the dependence on time. This method has since been explored in numerous empirical studies [17, 37, 43, 65, 57, 70, 4, 35]. However, despite these empirical successes, significant gaps

---

*Equal contribution. Department of Electrical and Computer Engineering, Princeton University. Authors' emails are: {yg6736, huiyuan, yy1325, minshuochen, mengdiw}@princeton.edu.

38th Conference on Neural Information Processing Systems (NeurIPS 2024).

remain in the theoretical understanding and guarantees of gradient-based guidance in diffusion models.

**Problem and Challenges** Suppose we have a pre-trained diffusion model that can generate new samples faithfully, maintaining the latent structure of data. We study the problem of adapting this diffusion model to generate new samples that optimize task-specific objectives, while *maintaining the learned structure* in new samples. This problem has a strong connection to classic optimization, guided diffusion offers new possibilities to optimize complex design variables such as images, videos, proteins, and genomes [7, 66, 42] in a generative fashion. More comprehensive exposure to this middle ground can be found in recent surveys [69, 12, 30].

Given the optimization nature of this problem, it's critical to answer the following theoretical questions from an optimization perspective: *(i)* Why doesn't simply applying the gradient of the objective function w.r.t. the noised sample work? *(ii)* How to add a guidance signal to improve the target objective without compromising the sample quality? *(iii)* Can one guarantee the optimization properties of new samples generated by guided diffusion? *(iv)* What are the limits of adaptability in these guided models?

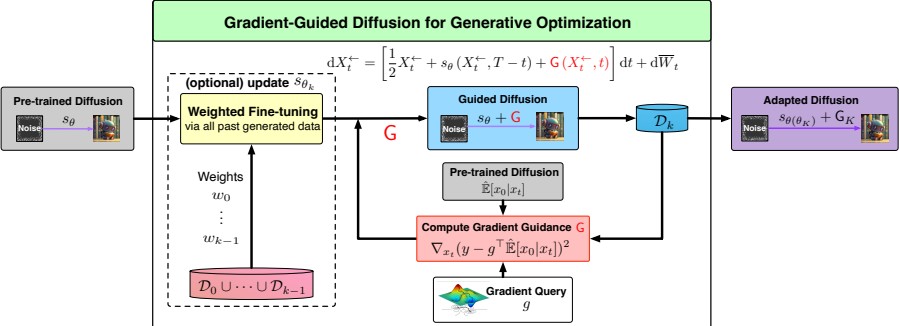

Figure 1: **Gradient-guided diffusion model for generative optimization, with or without adaptive fine-tuning.** A pre-trained diffusion model is guided with an additional gradient signal from an external objectives function towards generating near-optimal solutions.

**Scope and Contribution.** In this paper, we investigate guided diffusion from an optimization perspective. To answer the four questions above, we propose an algorithmic framework, see Figure 1 for an illustration. Our main contributions are summarized as follows:

• **Study structure-preserving guidance.** We study the design of guidance under structural data distribution belonging to a latent low-dimensional subspace (Assumption 1). We diagnose the failure of naive gradient guidance and study the theoretical aspects of guidance based on forward prediction loss (Definition 1), which provably preserves any low-dimensional subspace structure (Theorem 1).

• **Establish a mathematical framework of guided diffusion.** We build a mathematical framework for guided diffusion, which facilitates algorithm analysis and theory establishment. We propose and analyze an iterative guided diffusion using gradient queries on new samples (Algorithm 1; Figure 1 with fine-tuning block off). We give the first convergence theory showing generated samples converge to a regularized optimal solution w.r.t the objective (Theorem 4) with linear score class (12). The regularization is imposed by the pre-trained diffusion model, revealing a fundamental limit of adapting pre-trained diffusion models with guidance.

• **Provide rate-matching optimization theory.** Furthermore, we propose an adaptive gradient-guided diffusion, where both pre-trained score network and guidance are iteratively updated using self-generated samples (Algorithm 2; Figure 1 with the fine-tuning block turned on). We show in expectation its iteration converges to a global optima within the latent subspace, at a rate of $\tilde{\mathcal{O}}(1/K)$ (Theorems 3, $K$ is # of iterations), matching the classical convergence in convex optimization.

• **Provide experimental justification.** Simulation and image experiments are provided in Section 7 to support theoretical findings on latent structure-preserving and optimization convergence.

## 2 Related Works

To summarize the related work, we first give an overview of empirical studies relevant to our objective. We then discuss the theory of diffusion models, to which our main contribution is focused. Other

related topics, such as direct latent optimization in diffusion models and a detailed review of sampling and statistical theory of diffusion models, are deferred to Appendix A.

**Classifier Guidance and Training-free Guidance.**   [22] introduced classifier-based guidance, steering pre-trained diffusion models towards a particular class during inference. This method offers flexibility by avoiding task-specific fine-tuning, but still requires training a time-dependent classifier. Training-free guidance methods [16, 57, 70, 4, 32, 50, 28] eliminate the need for a time-dependent classifier, using only off-the-shelf loss guidance during inference. [16, 57, 32, 28] is a line of works solving inverse problems on image and [70, 50] aims for guided/conditional image generation. Though not originally developed for solving optimization problems, [16, 70] both propose a similar guidance to ours: taking gradient on the predicted clean data $x_0$ with respect to corrupted $x_t$. Differently, our paper presents the first rigorous theoretical analysis of this gradient-based guidance approach. Furthermore, we propose an algorithm that iteratively applies the guidance as a module to the local linearization of the optimization objective, demonstrating provable convergence guarantees.

**Fine-tuning of Diffusion Models.**   Several methods for fine-tuning diffusion models to optimize downstream reward functions include RL-based fine-tuning [7, 26] and direct backpropagation to rewards [18, 52, 68, 63]. However, these approaches often suffer from high computational costs and catastrophic forgetting in pre-trained models. Our guidance method is training-free and applied during the inference phase, eliminating the need to fine-tune diffusion models.

**Theory of Diffusion Models.**   Current theory works primarily focus on unconditional diffusion models. Several studies demonstrate that the distributions generated by diffusion models closely approximate the true data distribution, provided the score function is accurately estimated [20, 2, 8, 39, 13, 40, 15, 14, 6]. For conditional diffusion models, [71, 27] establish sample complexity bounds for learning generic conditional distributions. Our novel analysis establishes a connection between the sampling process in gradient-based guided diffusion and a proximal gradient step, providing convergence guarantees.

## 3   Preliminaries: Diffusion Models and Guidance

Score-based diffusion models capture the distribution of pre-training data by learning a sequence of transformations to generate new samples from noise [60]. A diffusion model comprises a forward and a backward process, for which we give a review as follows.

**Forward Process.**   The forward process progressively adds noise to data, and then the sample trajectories are used to train the score function. The forward process initializes with $X_0 \in \mathbb{R}^D$, a random variable drawn from the pre-training data $\mathcal{D}$. It introduces noise via an Ornstein-Uhlenbeck process, i.e.,

$$\mathrm{d}X_t = -\frac{1}{2}q(t)X_t \, \mathrm{d}t + \sqrt{q(t)} \, \mathrm{d}W_t \quad \text{for} \ \ q(t) > 0, \tag{1}$$

where $(W_t)_{t \geq 0}$ is Wiener process, and $q(t)$ is non-decreasing. $X_t$ represents the noise-corrupted data distribution at time $t$. The conditional distribution $X_t|X_0 = x_0$ is Gaussian, i.e., $\mathcal{N}(\alpha(t)x_0, h(t)I_D)$ with $\alpha(t) = \exp(-\int_0^t \frac{1}{2}q(s)ds)$ and $h(t) = 1 - \alpha^2(t)$. In practice, the forward process will terminate at a large time $T$ so that the marginal distribution of $X_T$ is close to $\mathcal{N}(0, I_D)$.

**Backward Process.**   If reversing the time of the forward process, we can reconstruct the original distribution of the data from pure noise. With $(\overline{W}_t)_{t \geq 0}$ being another independent Wiener process, the backward SDE below [3] reverses the time in the forward SDE (1),

$$\mathrm{d}X_t^{\leftarrow} = \left[\frac{1}{2}X_t^{\leftarrow} + \underbrace{\nabla \log p_{T-t}(X_t^{\leftarrow})}_{\text{score}}\right] \mathrm{d}t + \mathrm{d}\overline{W}_t. \tag{2}$$

Here $p_t(\cdot)$ denotes the marginal density of $X_t$ in the forward process. In the forward SDE (2), the *score function* $\nabla \log p_t(\cdot)$ plays a crucial role, but it has to be estimated from data.

**Score Matching.**   To learn the unknown score function $\nabla \log p_t(\cdot)$, we train a score network $s_\theta(x, t)$ using samples from forward process. Let $\mathcal{D}$ denote the data for training. Then the score network is

learned by minimizing the following loss:

$$\min_{s \in \mathcal{S}} \int_0^T \mathbb{E}_{x_0 \in \mathcal{D}} \mathbb{E}_{x_t | x_0} \left[ \| \nabla_{x_t} \log \phi_t(x_t | x_0) - s(x_t, t) \|^2 \right] \mathrm{d}t, \tag{3}$$

where $\mathcal{S}$ is a given function class, $\mathbb{E}_{\mathcal{D}}$ denotes the empirical expectation over training data $\mathcal{D}$ and $\mathbb{E}_{x_t | x_0}$ denotes condition expectation over the forward process, $\phi_t(x_t | x_0)$ is the Gaussian transition kernel, i.e., $(2\pi h(t))^{-D/2} \exp(-\|x_t - \alpha(t)x_0\|^2 / (2h(t)))$.

**Generation and Guided Generation.** Given a pre-trained score function $s_\theta$, one generates samples by the backward process (2) with the true score replaced by $s_\theta$. Further, one can add additional guidance to steer its output distribution towards specific properties, as formulated in Module 1.

---

**Module 1** `Guided_BackwardSample`$(s_\theta, \mathtt{G})$

---

1: **Input**: Score $s_\theta$, guidance $\mathtt{G}$ default to be zero for unguided generation.
2: **Hyper-parameter**: $T$.
3: Initialized at $X_t^\leftarrow \sim \mathcal{N}(0, I)$, simulate the following SDE till time $T$:

$$\mathrm{d}X_t^\leftarrow = \left[ \frac{1}{2} X_t^\leftarrow + s_\theta \left( X_t^\leftarrow, T - t \right) + \mathtt{G} \left( X_t^\leftarrow, T - t \right) \right] \mathrm{d}t + \mathrm{d}\overline{W}_t.$$

4: **Output**: Sample $X_T^\leftarrow$.

---

A common goal of guided generation (Module 1) is to generate $X$ with a desired property $Y = y$ from the distribution $P(X|Y = y)$. To this end, it essentially needs to learn the **conditional score function** $\nabla_{x_t} \log p_t(x_t \mid y)$. The Bayes rule gives

$$\nabla_{x_t} \log p_t(x_t \mid y) = \underbrace{\nabla \log p_t(x_t)}_{\text{est. by } s_\theta(x_t, t)} + \underbrace{\nabla_{x_t} \log p_t(y \mid x_t)}_{\text{to be est. by guidance}}. \tag{4}$$

When a pre-trained score network $s_\theta(x_t, t) \approx \nabla \log p_t(x_t)$, what remains is to estimate $\nabla_{x_t} \log p_t(y \mid x_t)$ and add it as the guidance term $\mathtt{G}$ to the backward process in Module 1.

**Classifier and Classifier-Free Guidance.** Classifier guidance [60, 22] samples from $P(X|Y = y)$ when $Y$ is a discrete label. This method estimates $\nabla_{x_t} \log p_t(y \mid x_t)$ by training auxiliary classifiers, denoted as $\hat{p}(y \mid x_t, t)$, and then computing the gradient of the classifier logits as the guidance, i.e., $\mathtt{G}(x_t, t) = \nabla_{x_t} \log \hat{p}(y \mid x_t, t)$. Alternatively, classifier-free guidance [33] jointly trains a conditional and an unconditional diffusion model, combining their score estimates to generate samples.

**Notations.** For a random variable $X$, $P_x$ represents its distribution, and $p(x)$ denotes its density. For $X, Y$ jointly distributed, $P(X \mid Y = y)$ denotes the conditional distribution, and $p(x \mid y)$ its density. The conditional expectation is denoted as $\mathbb{E}[x \mid y]$. Let $\mathcal{D}$ be the pre-training data, and let $\mathbb{E}_{\mathcal{D}}$ be the empirical expectation over $\mathcal{D}$. The empirical mean and covariance matrix are $\bar{\mu} := \mathbb{E}_{x \in \mathcal{D}}[x]$ and $\bar{\Sigma} := \mathbb{E}_{x \in \mathcal{D}}[(x - \bar{\mu})(x - \bar{\mu})^\top]$. For a matrix $A$, $\mathrm{Span}(A)$ denotes the subspace spanned by its column vectors, and for a square matrix $A$, $A^{-1}$ denotes its inverse or Moore–Penrose inverse. For any differentiable function $f : \mathbb{R}^n \to \mathbb{R}^m$, $\nabla f \in \mathbb{R}^{m \times n}$ denotes Jacobian matrix, i.e., $(\nabla f)_{ij} = \frac{\partial f_i(x)}{\partial x_j}$.

## 4 A Primer on Gradient Guidance

Let us start with stating the problem we want to study: suppose **given** a pre-trained diffusion model where the score network $s_\theta(x_t, t)$ provides a good approximation to the true score function $\nabla \log p(x_t)$, the **goal** is to generate novel samples with desired properties that can be measured by a user-specified differentiable function $f$. We will refer to $f$ as a reward or objective function later on. To achieve this goal, from Section 3, we know that guided generation with diffusion model is a good candidate, which deploys the following guided backward process (Module 1):

$$\mathrm{d}X_t^\leftarrow = \left[ \frac{1}{2} X_t^\leftarrow + s_\theta(X_t^\leftarrow, T - t) + \mathtt{G}(X_t^\leftarrow, t) \right] \mathrm{d}t + \mathrm{d}\overline{W}_t.$$

Here the guidance term $\mathtt{G}$ is what we focus on and wish to design. Specifically, we want to construct this guidance term $\mathtt{G}$ based on the gradient $\nabla f$ of a general objective $f$. This is motivated by the

gradient methodology in optimization, a natural, intuitive way for adding guidance is to steer the generated samples towards the steepest ascent direction of $f$ [16, 4, 18].

## 4.1 Structural Data Distribution with Subspace

When incorporating property optimization in the generation process, it's crucial to consider intrinsic low-dimensional structures of real-world data, such as local regularities, global symmetries, and repetitive patterns [62, 55, 51]. Blindly improving $f$ at the cost of losing these structures degrades sample quality dramatically. This quality degradation, also known as "reward over-optimization", is a common challenge for adapting diffusion models towards an external reward [71, 63].

To study the design of guidance that mitigates the risk of over-optimization, we focus on data that admits a low-dimensional latent subspace, formulated in the following assumption.

**Assumption 1** (Subspace Data). *Data $X \in \mathbb{R}^D$ can be represented as $X = AU$, where $A \in \mathbb{R}^{D \times d}$ is an unknown matrix with orthonormal columns, and the latent variable $U \in \mathbb{R}^d$ follows some distribution $P_u$ with a density $p_u$. Here $d \ll D$. The empirical covariance of $U$ is assumed full rank.*

In the rest of this section, we investigate the principles for designing a guidance based on the gradient of $f$ that ensures (i) improving the value of $f$, and at the same time, (ii) being adhere to the subspace structure, i.e. generated samples being close to the subspace spanned by $A$.

## 4.2 Naive Gradient Does't Work as Guidance

A tempting simple choice of the guidance G is by taking the steepest ascent direction $\nabla f$, which we refer to as *naive gradient guidance* i.e.,

$$\mathtt{G}(X_t^{\leftarrow}, t) \propto \nabla f(X_t^{\leftarrow}). \tag{5}$$

However, the naive gradient guidance (5) would jeopardize the latent structure of data, which is demonstrated by the following proposition:

**Proposition 1** (Failure of Naive Guidance). *For naive guidance $\mathtt{G}(X_t^{\leftarrow}, t) = b(t)\nabla f(X_t^{\leftarrow})$, suppose $b(t) > b_0 > 0$ for $t > t_0$. For data in subspace under Assumption 1 and reward $f(x) = g^\top x$, $g \perp \mathrm{Span}(A)$ with $h(t) = 1 - \exp(-\sqrt{t})$, then the off-subspace component of the generated sample is consistently large:*
$$\mathbb{E}[X_{T,\perp}^{\leftarrow}] = Cg, \quad C > \exp(-5/2)b_0.$$

The intuition provided by Proposition 1 is, while the pre-trained score network effectively steers the distribution toward the latent subspace [10], the gradient vector $\nabla f$ may point outside the subspace, causing the generated output to deviate from it (Figure 2). This is why naive gradient guidance fails. [63] also observed this, explaining that $\nabla f$ is not computed for $t = T$ i.e., it is not aligned with the clean data space.

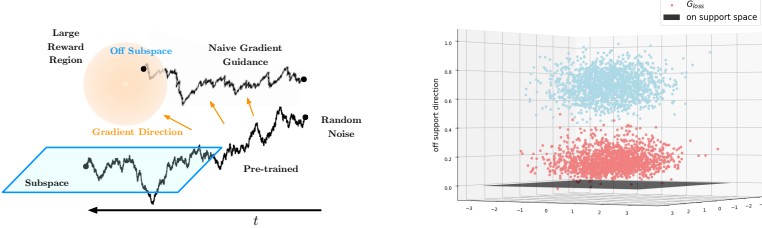

Figure 2: **Directly adding the gradient of the objective function to the backward process sabotages the subspace structure.** Left: Directly adding gradients that point out of the data subspace causes samples to leave the subspace. Right: Numerical experiments show that naive gradients lead to substantially larger off-subspace error compared to our gradient guidance $\mathtt{G}_{loss}$(Definition 1); see Section 7 for experiment details.

## 4.3 Motivating Latent Subspace Preserving Guidance from Conditional Score Function

Failure of the naive gradient in maintaining data structure motivates us to seek alternatives. To get some inspiration, we start with the most elementary Gaussian probabilistic model and linear $f$. Later we will drop this assumption and consider general data distributions and general $f$.

**Assumption 2** (Gaussian Linear model). *Let data follow a Gaussian distribution, i.e., $X \sim \mathcal{N}(\mu, \Sigma)$, and let $f(x) = g^\top x$ be a linear function for some $g \in \mathbb{R}^D$. Let $Y = f(X) + \epsilon$ with independent, identically distributed noise $\epsilon \sim \mathcal{N}(0, \sigma^2)$ for some $\sigma > 0$.*

By the Bayes' rule, the conditional score $\nabla_{x_t} \log p_t(x_t \mid y)$ takes the form of a sum given by

$$\nabla_{x_t} \log p_t(x_t \mid y) = \underbrace{\nabla \log p_t(x_t)}_{\text{est. by } s_\theta(x_t, t)} + \underbrace{\nabla_{x_t} \log p_t(y \mid x_t)}_{\text{to be est. by guidance}}. \qquad \text{(recall (4))}$$

Under the Gaussian assumption, we derive the following closed-form formula of the guidance term $\log p_t(y \mid x_t)$ that we want to estimate. The proof is provided in Appendix D.4.

**Lemma 1** (Conditional Score gives a Gradient-like Guidance). *Under Assumption 2, we have*

$$\nabla_{x_t} \log p_t(y \mid x_t) = - \left( 2\sigma_y^2(x_t) \right)^{-1} \cdot \nabla_{x_t} \left( y - g^\top \mathbb{E}[x_0 \mid x_t] \right)^2, \qquad (6)$$

*where $\mathbb{E}[x_0 | x_t]$ denotes the conditional expectation of $x_0$ given $x_t$ in the forward process (1), and $\sigma_y^2(x_t)$ is the variance of the conditional distribution $Y \mid X_t = x_t$.*

The form of conditional score shown in Lemma 1 motivates our proposed gradient guidance:

**Definition 1** (Gradient Guidance of Look-Ahead Loss). *Given a gradient vector g, define the gradient guidance of look-ahead loss as*

$$\mathsf{G}_{loss}(x_t, t) := -\beta(t) \cdot \nabla_{x_t} \left( y - g^\top \mathbb{E}[x_0 | x_t] \right)^2, \qquad (7)$$

*where $\beta(t) > 0, y \in \mathbb{R}$ are tuning parameters, and $\mathbb{E}[x_0 | x_t]$ is the conditional expectation of $x_0$ given $x_t$ in the forward process (1), i.e., $\mathrm{d}X_t = -\frac{1}{2} q(t) X_t \, \mathrm{d}t + \sqrt{q(t)} \, \mathrm{d}W_t$.*

The formula in (7) generalizes the intuition of a conditional score for any data distribution and objective function. It scales with the residual term $y - g^\top \mathbb{E}[x_0 \mid x_t]$, tuning the *strength of guidance*. Here, $\mathbb{E}[x_0 \mid x_t]$ represents the expected clean data $x_0$ given $x_t$ in the forward process, which coincides with the expected sample in the backward view. This residual measures the **look-ahead gap** between the expected reward of generated samples and the target value. The **look-ahead loss** $(y - g^\top \mathbb{E}[x_0 | x_t])^2$ resembles the proximal term commonly used in first-order proximal optimization methods.

**Remark.** The gradient guidance (7) aligns with the groundtruth conditional score in (6) under the assumptions of Gaussian data and linear reward (Assumption 2). This theoretical motivation, rooted in a fundamental framework, distinguishes our work from the empirical practice, such as DPS [16] and universal guidance [4].

A key advantage of $\mathsf{G}_{loss}$ is that it enables preserving the subspace structure, for **any** data distribution under Assumption 1. This is formalized in the following theorem, the full proof in Appendix D.5.

**Theorem 1** (Faithfulness of $\mathsf{G}_{loss}$ to the Low-Dimensional Subspace of Data). *Under Assumption 1, it holds for any data distribution and $g \in \mathbb{R}^D$ that*

$$\mathsf{G}_{loss}(x_t, t) \in \mathrm{Span}(A). \qquad (8)$$

**Proof Sketch.** We have

$$\nabla_{x_t} \left( y - g^\top \mathbb{E}[x_0 \mid x_t] \right)^2 \propto \nabla_{x_t} \mathbb{E}[x_0 \mid x_t]^\top g.$$

We will show that the Jacobian $\nabla_{x_t} \mathbb{E}[x_0 | x_t]$ maps any vector $g \in \mathbb{R}^D$ to $\mathrm{Span}(A)$. To see this, we utilize the score decomposition result in Appendix D.1 and plug it into the equality $\mathbb{E}[x_0 | x_t] = \alpha^{-1}(t) (x_t + h(t)\nabla \log p_t(x_t))$ (Tweedie's formula [24]), we have

$$\mathbb{E}[x_0 \mid x_t] = \alpha^{-1}(t) \left( x_t + h(t) \left[ Am(A^\top x_t) - h^{-1}(t)x_t \right] \right) = h(t)/\alpha(t) \cdot Am(A^\top x_t), \quad (9)$$

here $m(u) = \nabla \log p_t^{\mathsf{LD}}(u) + h^{-1}(t)u$, $p_t^{\mathsf{LD}}(u)$ latent density (Appendix D.1). We see $\nabla_{x_t} \mathbb{E}[x_0 | x_t]^\top$ maps any vector to $\mathrm{Span}(A)$ because $m(\cdot)$ takes $A^\top x_t$ as input in the expression of $\mathbb{E}[x_0 | x_t]$. ∎

We highlight that the faithfulness of $\mathsf{G}_{loss}$ holds for *arbitrary* data distribution supported on the latent subspace. It takes advantage of the score function's decomposition (19), having the effect of automatically adapting $g$ onto the latent low-dimensional subspace of data.

**Remark.** We provide a rigorous guarantee for manifold preservation of gradient guidance, a property previously discussed by [16, 17, 32]. However, while [16, 17] claim manifold preservation, they do not present a formal mathematical proof. [32] relies heavily on pre-trained autoencoders for manifold projections, which are often unavailable in practical scenarios.

## 4.4 Estimation and Implementation of $\mathtt{G}_{loss}$

In this section, we discuss the estimation and computation of $\mathtt{G}_{loss}$ based on a pre-trained score function $s_\theta$ in practice. $\mathtt{G}_{loss}$ involves the unknown quantity $\mathbb{E}[x_0|x_t]$. One can construct estimate $\mathbb{E}[x_0|x_t]$ by considering the Tweedie's formula [24]: $\nabla \log p_t(x_t) = -h^{-1}(t)\mathbb{E}\left[x_t - \alpha(t)x_0|x_t\right]$, which gives rise to

$$\hat{\mathbb{E}}[x_0|x_t] := \alpha^{-1}(t)\left(x_t + h(t)s_\theta(x_t,t)\right), \tag{10}$$

and we refer to it as the *look-ahead estimator*. The estimator (10) is widely adopted in practice [56, 4]. Here $\alpha(t)$ and $h(t)$ are the noise scheduling used in the forward process (1).

Thus, we have obtained an **implementable version of the gradient guidance $\mathtt{G}_{loss}$**, given by

$$\mathtt{G}_{loss}(x_t,t) = -\beta(t) \cdot \nabla_{x_t}\left[y - g^\top\left(\alpha^{-1}(t)\left(x_t + h(t)s_\theta(x_t,t)\right)\right)\right]^2, \tag{11}$$

With a slight abuse of notation, we use $\mathtt{G}_{loss}$ to refer to this implementable formula (11) in the remainder of this paper. Here, $y$ is a target reward value from conditional score analysis under a Gaussian model and is treated as a tuning parameter in practice. The gradient guidance (11) is lightweight to implement. Given a pre-trained score function $s_\theta$ in the form of a neural network, computing (11) involves calculating the squared loss $\left(y - g^\top\hat{\mathbb{E}}[x_0|x_t]\right)^2$ via a forward pass of $s_\theta$ and a backward pass using the auto-gradient feature of deep-leaning frameworks such as PyTorch and TensorFlow. See Figure 3 for illustration.

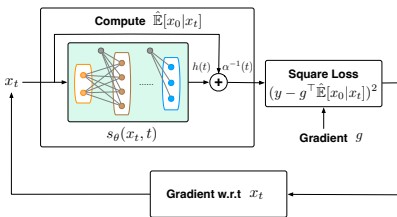

Figure 3: Computing $\mathtt{G}_{loss}$.

## 5 Gradient-Guided Diffusion Model as Regularized Optimizer

In this section, we study if gradient guidance steers pre-trained diffusion models to generate **near-optimal** samples. Our results show that: 1) Iterative gradient guidance improves the objective values; 2) The pre-trained diffusion model acts as a regularizer from an optimization perspective.

### 5.1 Gradient-Guided Generation with A Pre-trained Score

Assuming access to a pre-trained score network $s_\theta$ and the gradient of the objective function $f$, we present Algorithm 1 to adapt the diffusion model and iteratively update the gradient guidance (11). See Figure 1 for illustration.

Alg. 1 takes any pre-trained score function $s_\theta$ as input. Each iteration evaluates $\nabla f(\cdot)$ at samples from the previous iteration (Line 5(i)), computes the gradient guidance $\mathtt{G}_{loss}$ with the new gradient (Line 5(ii)), and generates new samples using the updated guidance (Module 1). The algorithm outputs an adapted diffusion model specified by $(s_\theta, \mathtt{G}_K)$.

---

**Algorithm 1** Gradient-Guided Diffusion for Generative Optimization

---

1: **Input**: Pre-trained score network $s_\theta(\cdot,\cdot)$, differentiable objective function $f$.
2: **Tuning Parameter**: Strength parameters $\beta(t)$, $\{y_k\}_{k=0}^{K-1}$, number of iterations $K$, batch sizes $\{B_k\}$.
3: **Initialization**: $\mathtt{G}_0 = $ NULL.
4: **for** $k = 0, \ldots, K-1$ **do**
5:     **Generate**: Sample $z_{k,i} \sim$ Guided_BackwardSample$(s_\theta, \mathtt{G}_k)$ using Module 1, for $i \in [B_k]$.
6:     **Compute Guidance**:
        (i) Compute the sample mean $\bar{z}_k := (1/B_k)\sum_{i=1}^{B_k} z_{k,i}$.
        (ii) Query gradient $g_k = \nabla f(\bar{z}_k)$.
        (iii) Update gradient guidance $\mathtt{G}_{k+1}(\cdot,\cdot) = \mathtt{G}_{loss}(\cdot,\cdot)$ via (7), using $s_\theta$, gradient vector $g_k$, and parameters $y_k$ and $\beta(t)$.
7: **end for**
8: **Output**: $(s_\theta, \mathtt{G}_K)$.

---

### 5.2 Gradient-Guided Diffusion Converges to Regularized Optima in Latent Space

We analyze the convergence of Alg. 1 and show that in final iterations, generated samples center around a regularized solution of the optimization objective $f$ within the subspace $\mathrm{Span}(A)$. Our theorems allow the pre-training data to have *arbitrary distribution*.

**Assumption 3** (Concave smooth objective). *The objective $f : \mathbb{R}^D \to \mathbb{R}$ is concave and $L$-smooth w.r.t. the (semi-)norm $\|\cdot\|_{\bar{\Sigma}^{-1}}$, i.e., $\|\nabla f(x_1) - \nabla f(x_2)\|_{\bar{\Sigma}} \leq L\|x_1 - x_2\|_{\bar{\Sigma}^{-1}}$ for any $x_1, x_2$.*

While Alg. 1 works with any pre-trained score network, we study its optimization properties focusing on the class of linear score functions given by

$$\mathcal{S} = \left\{ s(x,t) = C_t x + b_t : C_t \in \mathbb{R}^{D \times D}, \; b_t \in \mathbb{R}^D \right\}. \tag{12}$$

**Remark on the linear parametrization of score network** (12)**:** Analyzing the output distribution of guided diffusion is challenging because the additional guidance term destroys the dynamics of reverse SDE. A linear score is a natural and reasonable choice for characterizing the output distribution, as it was also adopted by [44].

With a linear score function (12), pre-training a diffusion model is equivalent to using a Gaussian model to estimate and sample from the estimated distribution. Thus, the guidance $\mathsf{G}_{loss}$ is also linear in $x_t$, and the final output follows a Gaussian distribution; see (25) in Appendix E. We focus on the mean, $\mu_K$, of the generated distribution from the backward sampling of $(s_\theta, \mathsf{G}_K)$ (as $T \to \infty$), and establish its optimization guarantee.

**Theorem 2** (Convergence to Regularized Maxima in Latent Subspace in Mean). *Let Assumptions 1 and 3 hold. Suppose we use the score function class* (12) *for pre-training and computing guidance. Then Alg.1 gives an adapted diffusion model that generates new samples that belong to* $\mathrm{Span}(A)$*. Further, for any* $\lambda > L$*, there exists* $\beta(t), \{y_k\}$ *and batch size* $B_k$*, such that with high probability* $1 - \delta$*, the mean of the output distribution* $\mu_K$ *converges to be near* $x^*_{A,\lambda}$*, and it holds*

$$f\left(x^*_{A,\lambda}\right) - f(\mu_K) = \lambda \left(\frac{L}{\lambda}\right)^K \mathcal{O}\left(d \log\left(\frac{K}{\delta}\right)\right),$$

*where* $x^*_{A,\lambda}$ *is an optimal solution of the regularized objective:*

$$x^*_{A,\lambda} = \underset{x \in \mathrm{Span}(A)}{\mathrm{argmax}} \left\{ f(x) - \frac{\lambda}{2} \|x - \bar{\mu}\|^2_{\bar{\Sigma}^{-1}} \right\}. \tag{13}$$

*where* $\bar{\mu}, \bar{\Sigma}$ *are empirical mean and covariance of pre-training data* $\mathcal{D}$*.*

**Remarks.** **(1)** The regularization term $\frac{\lambda}{2} \|x - \bar{\mu}\|^2_{\bar{\Sigma}^{-1}}$ (23) centers the data's mean $\bar{\mu}$ and is stronger in directions where the original data has low variance. Thus, the pre-trained score acts as a "prior" in the guided generation, favoring samples near the pre-training data, even with guidance.
**(2)** The regularization term cannot be arbitrarily small, as our theorem requires $\lambda \geq L$. Thus, only adding gradient guidance cannot achieve the global maxima. If the goal is global optima, the pre-trained score must be updated and refined with new data, as explored in Section 6.
**(3)** The convergence rate is linear in the latent dimension $d$, rather than data dimension $D$. Since $\mathsf{G}_{loss}$ is faithful to the latent subspace (Theorem 2), the generated samples and optimization iterates of Alg. 1 remain within $\mathrm{Span}(A)$. This leverage of the latent structure results in faster convergence.

# 6 Gradient-Guided Diffusion with Adaptive Fine-Tuning for Global Optimization

In the previous section, we have seen that adding guidance to a pre-trained diffusion model can't improve the objective function unlimitedly due to the pre-trained score function acting as a regularizer. We consider adaptively fine-tuning pre-trained diffusion models to attain global optima. Empirically, fine-tuning diffusion models using self-generated samples has been explored by [7, 18].

## 6.1 Adaptive Fine-Tuning Algorithm with Gradient Guidance

We propose an adaptive version of the gradient-guided diffusion, where both the guidance and the score are iteratively updated using self-generated samples. The full algorithm is given in Algorithm 2.

We introduce a weighting scheme to fine-tune the score network using a mixture of pre-training data and newly generated samples. In Round $k$, let $\mathcal{D}_1, \ldots, \mathcal{D}_k$ be sample batches generated from the previous rounds. Let $\{w_{k,i}\}_{i=0}^k$ be a set of weights. Conceptually, at Round $k$, we update the model by minimizing the weighted score matching loss:

$$\min_{s \in \mathcal{S}} \int_0^T \sum_{i=0}^k w_{k,i} \mathbb{E}_{x_0 \in \mathcal{D}_i} \mathbb{E}_{x_t | x_0} \left[ \|\nabla_{x_t} \log \phi_t(x_t | x_0) - s(x_t, t)\|_2^2 \right] \mathrm{d}t, \tag{14}$$

where $\mathcal{D}_0 := \mathcal{D}$ is the pre-training data. For illustration, please see also Figure 1, and the practical implementation of Alg. 2 is in Appendix F.

**Algorithm 2** Gradient-Guided Diffusion with **Adaptive Fine-tuning**

1: **Input**: Pre-trained score $s_\theta(\cdot,\cdot)$, differentiable objective function $f$.
2: **Tuning Parameter**: strength parameter $\beta(t)$, $\{y_k\}_{k=0}^{K-1}$, weights $\{\{w_{k,i}\}_{i=0}^{k}\}_{k=0}^{K-1}$, number of iterations $K$, batch sizes $\{B_k\}$.
3: **Initialize**: $s_{\theta_0} = s_\theta$, $\mathtt{G}_0 = \mathtt{NULL}$.
4: **for** $k = 0, \cdots, K-1$ **do**
5:     **Generate**: Sample a batch $\mathcal{D}_k = \{z_{k,i}\}_{i=1}^{B_k}$ from $\mathtt{Guided\_BackwardSample}(s_{\theta_k}, \mathtt{G}_k)$ (Module 1).
6:     **Compute Guidance:**
        (i) Compute sample mean $\bar{z}_k = (1/B_k)\sum_{i=1}^{B_k} z_{k,i}$, and query gradient $g_k = \nabla f(\bar{z}_k)$.
        (ii) Update $s_{\theta_k}$ to $s_{\theta_{k+1}}$ by minimizing the re-weighted objective (14).
        (iii) Compute $\mathtt{G}_{k+1}(\cdot,\cdot) = \mathtt{G}_{loss}(\cdot,\cdot)$ in (7), using $s_{\theta_{k+1}}$ and $g_k$, with parameter $y_k, \beta(t)$.
7: **end for**
8: **Output**: $(s_{\theta_K}, \mathtt{G}_K)$.

## 6.2 Guided Generation Finds Unregularized Global Optima

Finally, we analyze the optimization properties for gradient-guided diffusion model with iterative finetuning. We establish that the process of Algorithm 2 yields a final output distribution whose mean, denoted by $\mu_K$, converges to the global optimum of $f$.

For simplicity of analysis, we study the following function class

$$\mathcal{S}' = \left\{ s(x,t) = \hat{C}_t x + b_t : b_t \in \mathbb{R}^D \right\}, \tag{15}$$

where $\hat{C}_t$ is set to stay the same in pre-trained scores, with only $b_t$ updated during iterative fine-tuning.

**Theorem 3** (Convergence to Unregularized Maxima in Latent Subspace in Mean). *Let Assumptions 1 and 3 hold, and assume there exists $M > 0$ such that $\left\| x_{A,\lambda}^* \right\| < M$ for all $\lambda \geq 0$. Suppose we use the score function class (12) for pre-training $s_\theta$ and the class (15) for finetuning it. Then Algorithm 2 gives an adapted diffusion model that generates new samples belonging to $\mathrm{Span}(A)$. Further, there exists $\{\beta(t)\}, \{y_k\}, \{B_k\}$ and $\{w_{k,i}\}$, such that with probability $1-\delta$,*

$$f_A^* - f(\mu_K) = \mathcal{O}\left( \frac{dL^2 \log K}{K} \cdot \log\left(\frac{K}{\delta}\right) \right), \tag{16}$$

*where $f_A^* = \max\{f(x) | x \in \mathrm{Span}(A)\}$.*

Theorem 3 illustrates that fine-tuning a diffusion model with self-generated data can reach global optima while preserving the latent subspace structure. The convergence rate matches standard convex optimization in terms of gradient evaluations, $K$. Compared to standard gradient solvers, guided diffusion models leverage pre-training data to solve optimization problems in a low-dimensional space, preserving desired structures and enabling more efficient exploration and faster convergence.

# 7 Experiments

## 7.1 Simulation

We conduct numerical simulations of Algorithms 1 and 2, following the subspace setup described in Assumption 1. Specifically, we set $d = 16$, $D = 64$, The latent variable $u$ is drawn from $\mathcal{N}(0, I_d)$ and used to construct $x = Au$, where $A$ is a randomly generated orthonormal matrix. We define the objective function $f(x) = 10 - (\theta^\top x - 3)^2$. To approximate the score function, we employ a version of the U-Net [54] with 14.8M trainable parameters. More details including how to set up $\theta$ are provided in Appendix F.1.

**Preserving Subspace Structure.** We first demonstrate that $\mathtt{G}_{loss}$ preserves the subspace structure learned from the pre-trained model. For comparison, we also tested the naive guidance $\mathtt{G}(x_t, t) := \beta(t)\left(y - g^\top \mathbb{E}[x_0|x_t]\right) g$ (more details in Appendix F.1.). Figure 4 (left) shows that $\mathtt{G}_{loss}$ performs much better than the naive gradient $\mathtt{G}$ in preserving the linear subspace. Figure 4 (right)

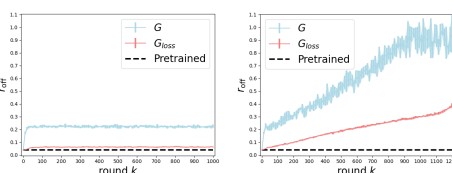

Figure 4: **Comparison between two types of gradient guidance** $\mathtt{G}$ **and** $\mathtt{G}_{loss}$ **(left: Alg. 1; right: Alg. 2).** The off/on support ratio of the generated samples is defined as $r_{\mathrm{off}} = \frac{\|x_\perp\|}{\|x_\|\|}$.

demonstrates that off-support errors increase with adaptive score fine-tuning (Alg. 2) due to distribution shift, with G resulting in more severe errors than $G_{loss}$.

**Convergence Results.** Figure 5 (a) and (b) show that Alg. 1 converges to a sub-optimal objective value, leaving a gap to the maximal value. This aligns with our theory that the pre-trained model acts as a regularizer in addition to the objective function. Figure 5 (c) shows that Alg 2 converges to the maximal value of the objective function. As illustrated by Figure 5 (d), samples from Alg. 1 mostly stay close to the pre-training data distribution (dotted contour area), whereas samples from Alg. 2 move outside the contour as the diffusion model is fine-tuned with self-generated data.

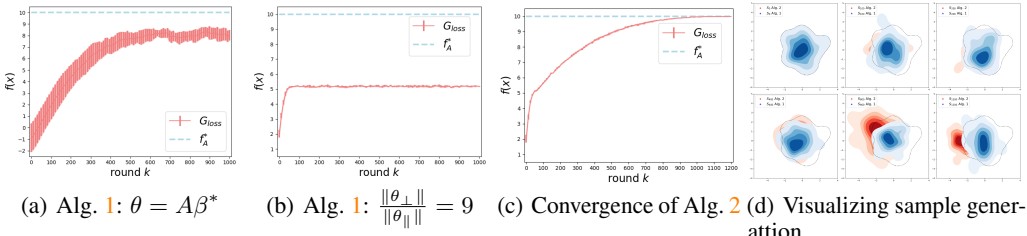

(a) Alg. 1: $\theta = A\beta^*$     (b) Alg. 1: $\frac{\|\theta_\perp\|}{\|\theta_\|\|} = 9$     (c) Convergence of Alg. 2     (d) Visualizing sample generation

Figure 5: **Convergence of Algorithms 1 and 2**. (a) and (b) are under different $\theta$ for the objective function. (d) visualizes the distribution of the generated samples of Alg. 2 (red) and Alg. 1 (blue) across the iterations.

## 7.2 Image Generation

We validate our theory in the image domain for Algorithm 1. We employ the StableDiffusion v1.5 model [53] as the pre-trained model. For the reward model, we follow the approach outlined by [71] to construct a synthetic model. This model is based on a ResNet-18 [31] architecture pre-trained on ImageNet [21], with the final prediction layer replaced by a randomly initialized linear layer that produces scalar outputs. For more experiment details, refer to Appendix F.2.

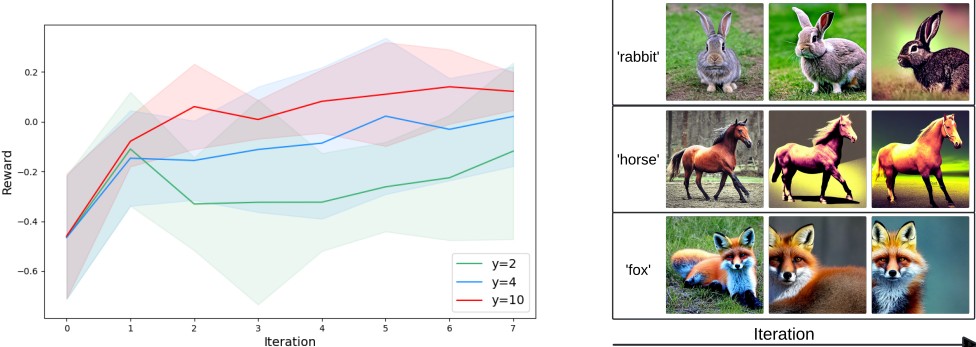

Figure 6: **Reward increase and effect on images across iterations.** Left: Reward increases and converges across iterations. Larger guidance strength $y$ (smaller regularizer strength) results in higher convergent reward value. Right: Images become more abstract, shifting from realistic to virtual backgrounds as reward increases.

**Results.** By Algorithm 1, the reward increases and converges. Figure 6 (left) shows the reward changes with optimization iterations. The hyperparameter $y$ tunes the strength of guidance and is inversely related to the strength of the regularizer (theoretical implications in Appendix E). A larger guidance strength (smaller regularizer strength) leads to a higher convergent reward value. Figure 6 (right) illustrates the changes in generated images across iterations. As the reward increases, the images become increasingly abstract, shifting from photo-realistic with detailed backgrounds to more virtual, stylized ones.

## 8 Conclusion

In this paper, we focus on gradient guidance for adapting or fine-tuning pre-trained diffusion models from an optimization perspective. We investigate the look-ahead loss based gradient guidance and two variants of diffusion-based generative optimization algorithms utilizing it. We provide guarantees for adapting/fine-tuning diffusion models to maximize any target concave differentiable reward function. Our analysis extends to linear subspace data, where our gradient guidance and adaptive algorithms preserve and leverage the latent subspace, achieving faster convergence to near-optimal solutions.

## Acknowledgments

Mengdi Wang acknowledges the support by NSF IIS-2107304, NSF CPS-2312093, ONR 1006977 and Genmab.

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

## A  Additional Related Works

**Direct Latent Optimization in Diffusion Models.**  Besides guidance methods, an alternative training-free route by optimizing the initial value of reverse process [64, 5, 36, 61, 49]. These methods typically backpropagate the gradient of reward directly to the initial latent vector through an ODE solver, utilizing the chain rule. Thus at inference time, the reverse process is unchanged except for being fed with an optimized initialization, different from the guidance method we studied.

**Sampling and Statistical Theory of Diffusion Model**  In contrast to the fruitful empirical advances, the theory of diffusion models is still limited. To the best of our knowledge, a theoretical understanding of fine-tuning diffusion models is absent. Existing results mainly focus on the sampling ability and statistical properties of unconditional diffusion models. In particular, for sampling ability, a line of works shows that the distribution generated by a diffusion model is close to the data distribution, as long as the score function is accurately estimated [20, 2, 8, 39, 13, 40]. The accuracy of the estimated score function is measured in terms of an $L_\infty$ or $L_2$-norm distance. More recently, [15, 14, 6] develop refined and tighter analyses using Taylor expansions of the discretized backward process and localization method. It is worth mentioning that the analysis in [15, 14, 6] extends to broader sample generation processes such as deterministic ones based on probabilistic ODEs. Going beyond distributions in Euclidean spaces, [19] analyzes diffusion models for sampling distribution supported on a low-dimensional manifold. Moreover, [46] consider sampling from symmetric spiked models, and [25] study sampling from Gibbs distributions using diffusion processes.

Turning towards the statistical theory of diffusion models, [59] and [41] provide asymptotic analyses, assuming a parametric form of the score function. Unfortunately, asymptotic analysis does not lead to concrete sample complexities. Later, concurrent works, [48] and [10], establish sample complexity bounds of diffusion models for estimating nonparametric data distributions. In high dimensions, their results highlight a curse of dimensionality issue without further assumptions, which also appears in [67] considering kernel methods. More interestingly, these works demonstrate that diffusion models can circumvent the curse of dimensionality issue if the data has low-dimensional structures. In the same spirit, [45] investigate learning high-dimensional graphical models using diffusion models, without the curse of dimensionality. [23] establishes statistical theory for consistency models. For conditional diffusion models, [71, 27] establish sample complexity bounds for learning generic conditional distributions. We refer readers to [12] for an overview of contemporary theoretical progress.

## B  Discussion

**Limitations**  We do not see significant limitations in our analysis. The linear score class initiates the study of gradient-like guidance. We have already demonstrated that our proposed gradient guidance preserves low-dimensional structures for any score and data distribution (Theorem 1). Moving forward, we aim to analyze the convergence properties for general score functions.

**Broader Impacts**  This paper's exploration of gradient-guided diffusion models offers theoretical implications that could inspire further algorithmic advancements. By establishing a connection between gradient-guided diffusion and optimization, the research opens new avenues for efficient and effective optimization through a generative approach. This work has the potential to enhance the precision and personalization of AI systems across various fields, including image synthesis and drug design. However, these techniques could also be misused to generate harmful content.

## C  Characterization for Output Distribution of Backward Process

In this section, we provide analytical characterizations for the output distribution of the backward process guided by $\mathsf{G}_{loss}$ when the pre-trained score is linear. We first give the result of score matching as follows.

**Lemma 2** (Pre-training with Linear Score Functions)**.**  *Suppose for pre-training the score network, the class in* (14) *is*

$$\mathcal{S} = \left\{ s(x,t) = C_t x + b_t : C_t \in \mathbb{R}^{D \times D}, \, b_t \in \mathbb{R}^D \right\}, \qquad \text{(recall (12))}$$

*If we freeze $C_t$ in* (12), *that is, minimizing the score matching objective* (14) *over the class* $\left\{ s(x,t) = C_t x + b_t : b_t \in \mathbb{R}^D \right\}$ *gives*

$$s_\theta(x_t, t) = C_t \left( x_t - \alpha(t)\bar{x} \right),$$

*where* $\bar{x} = \left( \sum_{i=0}^{k} w_{k,i} \mathbb{E}_{x \in \mathcal{D}_i}[x] \right) \Big/ \left( \sum_{i=0}^{k} w_{k,i} \right)$. *Moreover, minimizing the score matching objective* (14) *over the class* (12) *yields*

$$s_\theta(x_t, t) = - \left( \alpha^2(t)\bar{\Sigma} + h(t)I_D \right)^{-1} \left( x_t - \alpha(t)\bar{x} \right),$$

*where* $\bar{\Sigma} = \left( \sum_{i=0}^{k} w_{k,i} \mathbb{E}_{x \in \mathcal{D}_k} \left[ (x - \bar{x})(x - \bar{x})^\top \right] \right) \Big/ \left( \sum_{i=0}^{k} w_{k,i} \right)$ *are weighted data covariance.*

*Proof.* Using the linear score network class $\mathcal{S}$ with freezing $C_t$, we cast the score matching loss (14) into

$$\int_0^T \sum_{i=0}^{k} w_{k,i} \mathbb{E}_{x_0 \in \mathcal{D}_i} \mathbb{E}_{x_t \sim \mathcal{N}(\alpha(t)x_0, h(t)I_D)} \left[ \left\| \nabla_{x_t} \log \phi_t(x_t | x_0) - s(x_t, t) \right\|^2 \right] dt$$

$$= \int_0^T \sum_{i=0}^{k} w_{k,i} \mathbb{E}_{x_0 \in \mathcal{D}_i} \mathbb{E}_{x_t \sim \mathcal{N}(\alpha(t)x_0, h(t)I_D)} \left[ \left\| -\frac{1}{h(t)}(x_t - \alpha(t)x_0) - C_t x_t - b_t \right\|^2 \right] dt$$

$$= \int_0^T \sum_{i=0}^{k} w_{k,i} \mathbb{E}_{x_0 \in \mathcal{D}_i} \mathbb{E}_{x_t \sim \mathcal{N}(\alpha(t)x_0, h(t)I_D)} \left[ \left\| \left(C_t + \frac{1}{h(t)}I_D\right)(x_t - \alpha(t)x_0) + (\alpha(t)C_t x_0 + b_t) \right\|^2 \right] dt$$

$$\overset{(i)}{=} \int_0^T \sum_{i=0}^{k} w_{k,i} \mathbb{E}_{x_0 \in \mathcal{D}_i} \left[ \left\| b_t + \alpha(t)C_t x_0 \right\|^2 \right] + w \int_0^T \text{trace}\left( h(t)\left(C_t + \frac{1}{h(t)}I_D\right)^\top \left(C_t + \frac{1}{h(t)}I_D\right) \right) dt,$$

where $w = \sum_{i=0}^{k} w_{k,i}$, equality $(i)$ follows from computing the expectation over the conditional Gaussian distribution of $x_t | x_0$. We note that $b_t$ should minimize $\sum_{i=0}^{k} w_{k,i} \mathbb{E}_{x_0 \in \mathcal{D}_i} \left[ \left\| b_t + \alpha(t)C_t x_0 \right\|^2 \right]$ for any $t$, which leads to

$$\hat{b}_t = -\alpha(t)C_t \bar{x}.$$

Now, we solve $C_t$ for the second result. Substituting $\hat{b}_t$ into the optimization objective (14) yields:

$$\int_0^T \sum_{i=0}^{k} w_{k,i} \mathbb{E}_{x_0 \in \mathcal{D}_i} \left[ \left\| \alpha(t)C_t \bar{x} - \alpha(t)C_t x_0 \right\|^2 \right] + w \int_0^T \text{trace}\left( h(t)\left(C_t + \frac{1}{h(t)}I_D\right)^\top \left(C_t + \frac{1}{h(t)}I_D\right) \right) dt$$

$$= \int_0^T \alpha^2(t) \sum_{i=0}^{k} w_{k,i} \mathbb{E}_{x_0 \in \mathcal{D}_i} \left[ \left\| C_t(x_0 - \bar{x}) \right\|^2 \right] + w \int_0^T \text{trace}\left( h(t)\left( C_t^\top C_t + \frac{1}{h(t)}C_t + \frac{1}{h(t)}C_t^\top + \frac{1}{h^2(t)}I_D \right) \right) dt.$$

Taking the gradient for $C_t$, we get

$$2\alpha^2(t)C_t \sum_{i=0}^{k} w_{k,i} \mathbb{E}_{x_0 \in \mathcal{D}_i} \left[ (x_0 - \bar{x})(x_0 - \bar{x})^\top \right] + 2wh(t)C_t + 2wI_D.$$

Setting the gradient above to 0, we get the solution for $C_t$ as

$$\hat{C}_t = - \left( \alpha^2(t)w^{-1} \sum_{i=0}^{k} w_{k,i} \mathbb{E}_{x_0 \in \mathcal{D}_i} \left[ (x_0 - \bar{x})(x_0 - \bar{x})^\top \right] + h(t)I_D \right)^{-1}.$$

Therefore, the proof is completed.

$\square$

When $w_{k,0} = 1, w_{k,i} = 0, i \in [k]$, Lemma 2 reduces to the pre-traning score matching.

**Corollary 1.** *Let $\mathcal{D}$ be the pre-training data. Minimizing the score matching objective* (3) *over the function class* (12) *gives*

$$s_\theta(x_t, t) = - \left( \alpha^2(t)\bar{\Sigma} + h(t)I_D \right)^{-1} \left( x_t - \alpha(t)\bar{\mu} \right). \tag{17}$$

The following lemma characterizes the output distribution of the backward process guided by $\mathsf{G}_{loss}$ when the pre-trained score is linear.

**Lemma 3.** *If the pre-trained score $s_\theta(x_t, t)$ is (17), substituting the score function with $s_\theta(x_t, t) + \mathsf{G}_{loss}(x_t, t)$ in the backward SDE (2) yields, when $T \to \infty$,*

$$X_T^\leftarrow \stackrel{d}{=} \mathcal{N}\left(\bar{\mu} + \frac{y - g^\top \mu}{\sigma^2 + g^\top \bar{\Sigma} g} \bar{\Sigma} g, \bar{\Sigma} - \frac{\bar{\Sigma} g g^\top \bar{\Sigma}}{\sigma^2 + g^\top \bar{\Sigma} g}\right).$$

*with $\beta(t)$ assigned as $\beta(t) = \frac{1}{2}\left(\sigma^2 + g^\top \bar{\Sigma}^{-1}\left(I_D + \alpha^2(t)\bar{\Sigma}/h(t)\right)^{-1} g\right)^{-1}$. Moreover, if pre-training data reside in $\mathrm{Span}(A)$ following Assumption 1, it holds $X_T^\leftarrow \in \mathrm{Span}(A)$.*

*Proof.* Consider $X_0 \stackrel{d}{=} \mathcal{N}\left(\bar{\mu}, \bar{\Sigma}\right), Y = g^\top X_0 + \epsilon$ where $\epsilon \sim \mathcal{N}\left(0, \sigma^2\right)$. Let $X_0$ be the initialization of the forward process. Similar to the proof in Appendix D.4, we get

$$\begin{bmatrix} X_t \\ Y \end{bmatrix} \stackrel{d}{=} \mathcal{N}\left(\begin{bmatrix} \alpha(t)\bar{\mu} \\ g^\top \bar{\mu} \end{bmatrix}, \begin{bmatrix} \alpha^2(t)\bar{\Sigma} + h(t)I_D & \alpha(t)\bar{\Sigma} g \\ \alpha(t)g^\top \bar{\Sigma} & \sigma^2 + g^\top \bar{\Sigma} g \end{bmatrix}\right).$$

Thus, we get $s_\theta(x_t, t)$ is exactly the score of marginal distribution of $X_t$, i.e., $\nabla \log p_t(x_t) = s_\theta(x_t, t)$. According to the proof in Appendix D.4, we get $s_\theta(x_t, t) + \mathsf{G}_{loss}(x_t, t) = \nabla \log p_t(x_t \mid y)$. Thus, the backward SDE turns out to be

$$\mathrm{d}X_t^\leftarrow = \left[\frac{1}{2}X_t^\leftarrow + \nabla \log p_{T-t}(X_t^\leftarrow \mid y)\right]\mathrm{d}t + \mathrm{d}\overline{W}_t, \quad X_0^\leftarrow \stackrel{d}{=} \mathcal{N}\left(0, I_D\right). \quad (18)$$

The initial distribution $p_0(x_0 \mid y)$ of the forward process can also be obtained by (18) where we replace the initial distribution as $p_T(x_T \mid y)$. According to the data processing inequality, we get the bound of the total variation distance between the terminal distribution $p_T^\leftarrow$ of (18) and $p_0(x_0 \mid y)$:

$$\mathrm{TV}\left(p_0, p_T^\leftarrow\right) \leq \mathrm{TV}\left(p_T, \varphi\right),$$

where $p_0, p_T$ are short hands for $p_0(x_0 \mid y)$ and $p_T(x_T \mid y)$, and $\varphi(\cdot)$ is the density for the standard normal distribution $\mathcal{N}(0, I_D)$. Since in the forward process, $p_T \to \varphi$ when $T \to \infty$, we have $\mathrm{TV}\left(p_0, p_T^\leftarrow\right) \to 0$ when $T \to \infty$. We complete the first part of the lemma. As for the second part, if data reside in $\mathrm{Span}(A)$ following Assumption 1, we have $\bar{\mu} = A\bar{u}$ and $\bar{\Sigma} = A\bar{\Sigma}_u A^\top$, where $\bar{u} = \mathbb{E}_{x \in \mathcal{D}, u = A^\top x}[u]$, $\bar{\Sigma}_u = \mathbb{E}_{x \in \mathcal{D}, u = A^\top x}[(u - \bar{u})(u - \bar{u})^\top]$. Thus, the covariance matrix of $X_T^\leftarrow$ is

$$\bar{\Sigma} - \frac{\bar{\Sigma} g g^\top \bar{\Sigma}}{\sigma^2 + g^\top \bar{\Sigma} g} = A\left[\bar{\Sigma}_u - \frac{\bar{\Sigma}_u A^\top g g^\top A \bar{\Sigma}_u}{\sigma^2 + g^\top \bar{\Sigma} g}\right]A^\top,$$

and due to $X_T^\leftarrow$ follows Gaussian distribution, we get $X_T^\leftarrow \in \mathrm{Span}(A)$. Thus, the proof is completed. $\qquad \square$

# D  Additional Materials for Section 4

**Contents**

## D.1  Score decomposition for subspace data

Under Assumption 1, the score function $\nabla \log p_t(x)$ decomposes to two orthogonal parts: an on-support component belonging to the subspace; and an orthogonal component. We recall this key result in Proposition 2, which later plays a key role in deriving subspace preserving guidance.

**Proposition 2** (Score Decomposition for Subspace Data ([10] Lem. 1, Thm. 3))**.** *Under Assumption 1, the score function $\nabla \log p_t(x)$ decomposes as*

$$\nabla \log p_t(x) = \underbrace{A\nabla \log p_t^{\mathsf{LD}}(A^\top x)}_{\mathbf{s}_\parallel(A^\top x, t): \text{ on-support score}} - \underbrace{h^{-1}(t)\left(I_D - AA^\top\right)x}_{\mathbf{s}_\perp(x, t): \text{ ortho. score}}. \quad (19)$$

*where $p_t^{\mathsf{LD}}(u') = \int \phi_t(u' \mid u) p_u(u)\, \mathrm{d}u$ with $\phi_t(\cdot \mid u)$ being the density of $\mathcal{N}(\alpha(t)u, h(t)I_d)$ for the same $\alpha(t)$ and $h(t)$ in the forward process (1).*

### D.2 Proof of Proposition 1

We give the proof of Proposition 1, which shows the failure of naive gradient guidance.

*Proof.* Under Assumption 1, the score can be decomposed to terms parallel and orthogonal to $\mathrm{Span}(A)$ (Proposition 2). Applying naive guidance, we examine the orthogonal reverse process:

$$\mathrm{d}X^{\leftarrow}_{t,\perp} = \left[\frac{1}{2} - \frac{1}{h(T-t)}\right] X^{\leftarrow}_{t,\perp}\mathrm{d}t + b(t)g\mathrm{d}t + \left(I_D - AA^\top\right)\mathrm{d}\overline{W}_t.$$

Solving this SDE, we get the expectation of the final state following $\mathbb{E}[X^{\leftarrow}_{T,\perp}] = \int_0^T \exp\left(-\int_0^t h^{-1}(s)\mathrm{d}s\right) e^{t/2} b(T-t)g\mathrm{d}t$. For the schedule $h(t) = 1 - \exp(-\sqrt{t})$, we have the coefficient of direction $g$ is larger than $\int_0^T \exp(-t/2 - 2\sqrt{t})b(T-t)\mathrm{d}t > \int_0^1 \exp(-5/2)b_0\mathrm{d}t > 0$ where we can assume $T > 1$. Thus, $\mathbb{E}[X^{\leftarrow}_{T,\perp}] \neq 0$. This means the generated sample is leaving the subspace, i.e., naive gradient guidance will violate the latent structure. $\square$

### D.3 Discussion for gradient-like guidance

To further clarify the derivation of our gradient guidance, we present a variant of Lemma 1.

**Proposition 3.** *Under Assumption 2, we have*

$$\nabla_{x_t} \log p_t(y|x_t) = \beta(t)\left[y - g^\top\mathbb{E}[x_0|x_t]\right] \cdot \left(\alpha^2(t)\Sigma + h(t)I_D\right)^{-1}\Sigma g, \qquad (20)$$

*where $\mathbb{E}[x_0|x_t]$ denotes the conditional expectation of $x_0$ given $x_t$ in the forward process (1), and $\beta(t) = \alpha(t)/(\sigma^2 + g^\top\Sigma^{-1}\left(I_D + \alpha^2(t)/h(t)\cdot\Sigma\right)^{-1}g)$.*

**Remarks.** Observe that, when $\Sigma = I$, (20) suggests the following form of guidance that is aligned with the naive gradient, i.e., the steepest ascent direction:

$$\mathsf{G}(x_t, t) \propto \left[y - g^\top\mathbb{E}[x_0|x_t]\right]\cdot g.$$

However, even for Gaussian distributions, as long as $\Sigma \neq I$, the term of (6) is no longer proportional to $g$ but becomes a pre-conditioned version of the gradient. We show the guidance above can maintain the subspace structure of data in the experiments Appendix F.

Another observation is that this guidance scales with a residual term $y - g^\top\mathbb{E}[x_0 \mid x_t]$. In particular, the residual term $y - g^\top\mathbb{E}[x_0 \mid x_t]$ tunes the *strength of guidance*. Recall $\mathbb{E}[x_0 \mid x_t]$ denotes the posterior expectation of clean data $x_0$ given $x_t$ in the forward process. Thus, in a backward view, $\mathbb{E}[x_0 \mid x_t]$ coincides with the expected sample to be generated conditioned on $x_t$. In this sense, the quantity $y - g^\top\mathbb{E}[x_0 \mid x_t]$ measures a **look-ahead gap** between the expected reward of generated samples and the target value. A larger absolute value of the residual means stronger guidance in the backward generation process.

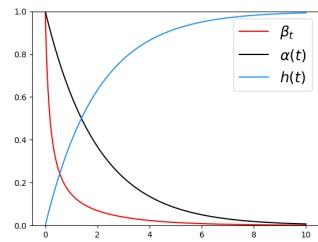

Figure 7: **Plot of** $\beta(t), \alpha(t), h(t)$ **for** $t \in [0, 10]$ when $\Sigma = I$.

Under $q(t) \equiv 1$ in the forward process (1), we plot the theoretical choice of $\beta(t)$ and $\alpha(t), h(t)$ to $t$ in Figure 7. In practice, the choice of $\alpha(t), h(t)$ can vary and they are determined by the forward process used for pre-training; and $\beta(t)$ can be treated as a tuning parameter to adjust the strength of guidance.

The proof of Proposition 3 is included in the proof of Lemma 1.

### D.4 Proof of Lemma 1

*Proof.* Recall $\{X_t\}_{t\geq 0}$ is the stochastic process from the forward process. $X_0 \overset{d}{=} \mathcal{N}(\mu, \Sigma)$, $Y = g^\top X_0 + \epsilon$ where $\epsilon \sim \mathcal{N}(0, \sigma^2)$ is independent with $X_0$. Since $X_0, X_t$ and $Y$ are joint Gaussian, we have $Y \mid X_t$ also follows Gaussian distribution, denoted as $\mathcal{N}\left(m_y(x_t), \sigma_y^2(x_t)\right)$. Then, the closed form of $\nabla_{x_t} \log p_t(y \mid x_t)$ can be derived as

$$\nabla_{x_t} \log p_t(y \mid x_t) = -\nabla_{x_t}\left[\frac{1}{2}\left(\frac{y - m_y(x_t)}{\sigma_y(x_t)}\right)^2\right] - \nabla_{x_t} \log \sigma_y(x_t).$$

Due to the linearity of $Y$ with regard to $X_0$, $m_y(x_t)$ can be computed as

$$m_y(x_t) = \mathbb{E}[y \mid x_t] = \mathbb{E}[g^\top x_0 + \epsilon \mid x_t] = \mathbb{E}[g^\top x_0 \mid x_t] = g^\top \mathbb{E}[x_0 \mid x_t]. \qquad (21)$$

To get the variance $\sigma_y^2(x_t)$, we compute the joint distribution $(X_t, Y)$. In the forward process, given $X_0 = x_0$, $X_t$ can be written as $\alpha(t)x_0 + Z_t$ for $Z_t \stackrel{d}{=} \mathcal{N}(0, h(t)I_D)$ independent of $x_0$. Due to the linear function assumption, we have

$$\begin{bmatrix} X_t \\ Y \end{bmatrix} = \begin{bmatrix} \alpha(t)I_D & 0 & I_D \\ g^\top & 1 & 0 \end{bmatrix} \begin{bmatrix} x_0 \\ \epsilon \\ Z_t \end{bmatrix}.$$

Observing that the joint distribution of $(x_0, \epsilon, z_t)$ is Gaussian, we deduce

$$\begin{bmatrix} X_t \\ Y \end{bmatrix} \stackrel{d}{=} \mathcal{N}\left( \begin{bmatrix} \alpha(t)\mu \\ g^\top \mu \end{bmatrix}, \begin{bmatrix} \alpha^2(t)\Sigma + h(t)I_D & \alpha(t)\Sigma g \\ \alpha(t)g^\top \Sigma & \sigma^2 + g^\top \Sigma g \end{bmatrix} \right).$$

Thus, we get $\sigma_y^2(x_t) = \sigma^2 + g^\top \Sigma g - \alpha^2(t)g^\top \Sigma \left(\alpha^2(t)\Sigma + h(t)I_D\right)^{-1} \Sigma g$. Together with the derivation of the mean $m_y(x_t)$ (21), we get

$$\nabla_{x_t} \log p_t(y \mid x_t) = -\frac{1}{2\sigma_y^2(x_t)} \nabla_{x_t} \left[ \left(y - g^\top \mathbb{E}[x_0 \mid x_t]\right)^2 \right]$$

$$= \frac{1}{\sigma_y^2(x_t)} \left(y - g^\top \mathbb{E}[x_0 \mid x_t]\right) \nabla_{x_t} \mathbb{E}[x_0 \mid x_t]g.$$

To get $\mathbb{E}[x_0 \mid x_t]$, we derive the joint distribution of $(X_0, X_t)$:

$$\begin{bmatrix} X_0 \\ X_t \end{bmatrix} \stackrel{d}{=} \mathcal{N}\left( \begin{bmatrix} \mu \\ \alpha(t)\mu \end{bmatrix}, \begin{bmatrix} \Sigma & \alpha(t)\Sigma \\ \alpha(t)\Sigma & \alpha^2(t)\Sigma + h(t)I_D \end{bmatrix} \right).$$

Thus, we get $\mathbb{E}[x_0|x_t] = \mu + \alpha(t) \left(\alpha^2(t)\Sigma + h(t)I_D\right)^{-1} \Sigma \left(x_t - \alpha(t)\mu\right)$. As a consequence, we have

$$\nabla_{x_t} \log p_t(y \mid x_t) = \frac{1}{\sigma_y^2(x_t)} \left(y - g^\top \mathbb{E}[x_0 \mid x_t]\right) \alpha(t) \left(\alpha^2(t)\Sigma + h(t)I_D\right)^{-1} \Sigma g.$$

Together with the following equality by Woodbury identity, we get the result.

$$\sigma_y^{-2}(x_t) = \left( \sigma^2 + g^\top \Sigma g - \alpha^2(t)g^\top \Sigma \left(\alpha^2(t)\Sigma + h(t)I_D\right)^{-1} \Sigma g \right)^{-1}$$

$$= \left[ \sigma^2 + g^\top \Sigma^{-1} \left(I_D + \frac{\alpha^2(t)}{h(t)}\Sigma\right)^{-1} g \right]^{-1}.$$

$\square$

### D.5 Proof of Theorem 1

*Proof.* We expand the derivative in $\mathsf{G}_{loss}$ as

$$\mathsf{G}_{loss}(x_t, t) = 2\beta(t)(y - g^\top \mathbb{E}[x_0|x_t]) \left(\nabla_{x_t} \mathbb{E}[x_0|x_t]\right)^\top g.$$

It holds $\mathbb{E}[x_0|x_t] = \frac{1}{\alpha(t)}(x_t + h(t)\nabla \log p_t(x_t))$. Via the score decomposition under linear subspace data in Chen et al. [10, Lemma 1], we have

$$\nabla \log p_t(x_t) = A\nabla \log p_t^{\mathsf{LD}}(A^\top x_t) - \frac{1}{h(t)}\left(I_D - AA^\top\right)x_t,$$

where $p_t^{\mathsf{LD}}$ denotes the diffused latent distribution, i.e., $p_t^{\mathsf{LD}}(u') = \int \phi_t(u'|u_0)p_u(u_0)\,\mathrm{d}u_0$. Recall that $\phi_t$ is the Gaussian transition kernel of the forward process and $p_u$ is the density of latent variable $u_0$ in Assumption 1.

To ease the derivation, we denote $m(u) = \nabla \log p_t^{\mathsf{LD}}(u) + \frac{1}{h(t)}u$. It then holds that

$$\mathbb{E}[x_0|x_t] = \frac{1}{\alpha(t)} \left( x_t + h(t) \left[ Am(A^\top x_t) - \frac{1}{h(t)}x_t \right] \right)$$

$$= \frac{h(t)}{\alpha(t)} Am(A^\top x_t).$$

As a consequence, we can verify that

$$\nabla_{x_t} \mathbb{E}[x_0|x_t] = \frac{h(t)}{\alpha(t)} A \left[ \nabla m(A^\top x_t) \right] A^\top,$$

where $\nabla m(A^\top x_t) \in \mathbb{R}^{d \times d}$ is the Jacobian matrix of $m$ at $A^\top x_t$. Plugging the last display into $\mathsf{G}_{loss}$, we conclude that

$$\mathsf{G}_{loss}(x_t, t) = 2\beta(t)(y - g^\top \mathbb{E}[x_0|x_t]) \left( \nabla_{x_t} \mathbb{E}[x_0|x_t] \right)^\top g$$
$$= \ell_t \cdot g$$

for $\ell_t = 2\beta(t)(y - g^\top \mathbb{E}[x_0|x_t])$ and $g' = \left( \nabla_{x_t} \mathbb{E}[x_0|x_t] \right)^\top g = \frac{h(t)}{\alpha(t)} A \left[ \nabla m(A^\top x_t) \right]^\top A^\top g \in$ Span$(A)$. The proof is complete. $\qquad \square$

# E  Additional Materials for in Sections 5 and 6

**Contents**

## E.1  Convergence to Regularized Maxima in Mean

Here we present a more fundamental version of Theorem 2.

**Theorem 4** (Convergence to Regularized Maxima in Mean). *Let Assumption 3 hold, and let the pre-training data $\mathcal{D}$ have arbitrary distribution with covariance matrix $\bar{\Sigma} \succ 0$. Suppose the score function $s_\theta$ is pre-trained via minimizing the score matching loss (3) over the linear function class (12). Let Alg. 1 take $s_\theta(\cdot, \cdot)$ and $f$ as the input. For any $\lambda > L$, there exists $\{\beta(t)\}$, $\{y_k\}$, $\{B_k\}$ such that, with probability $\geq 1 - \delta$, the mean of the output distribution $\mu_K$ converges to be near $x_\lambda^*$, and*

$$f(x_\lambda^*) - f(\mu_K) = \lambda \left( \frac{L}{\lambda} \right)^K \mathcal{O} \left( D \log \left( \frac{K}{\delta} \right) \right), \tag{22}$$

*where $D$ is the ambient dimension of data, and $x_\lambda^*$ is a regularized maximizer of $f$ given by*

$$x_\lambda^* = \underset{x \in \mathbb{R}^D}{\arg\max} \left\{ f(x) - \frac{\lambda}{2} \|x - \bar{\mu}\|_{\bar{\Sigma}^{-1}}^2 \right\}, \tag{23}$$

*where $\bar{\mu}, \bar{\Sigma}$ are empirical mean and covariance of pre-training data $\mathcal{D}$.*

*Proof.* The proof is a special case of Theorem 2 in Appendix E.2, via setting the representation matrix $A = I_D$. $\qquad \square$

## E.2  Proof of Theorem 2

We first provide a proof sketch.

**Proof Sketch**   Solving the score matching problem (3) with a linear function class (12) yields a pre-trained score as follows

$$s_\theta(x_t, t) = - \left( \alpha^2(t)\bar{\Sigma} + h(t)I_D \right)^{-1} (x_t - \alpha(t)\bar{\mu}).$$

With proper choices of $\beta(t)$, gradient guidance $\mathsf{G}_{loss}$ leads to the following output distribution at the end of round $k$:

$$\mathcal{N} \left( \bar{\mu} + \frac{y_k - g_k^\top \bar{\mu}}{\sigma^2 + g_k^\top \bar{\Sigma} g_k} \bar{\Sigma} g_k, \bar{\Sigma} - \frac{\bar{\Sigma} g_k g_k^\top \bar{\Sigma}}{\sigma^2 + g_k^\top \bar{\Sigma} g_k} \right).$$

Thus, we obtain the mean of the above distribution, i.e., $\mu_{k+1} = \bar{\mu} + \eta_k \bar{\Sigma} \nabla f(\bar{z}_k)$, where $\bar{z}_k$ is the empirical mean of previous samples, $\eta_k$ is a stepsize determined by $y_k$. By a rearrangement, we obtain a recursive formula

$$\mu_{k+1} = \bar{z}_k + \eta_k \bar{\Sigma} \left[ \nabla f(\bar{z}_k) - \eta_k^{-1} \bar{\Sigma}^{-1} (\bar{z}_k - \bar{\mu}) \right]. \tag{24}$$

We observe that (24) resembles a gradient ascent update from $\mu_k \approx \bar{z}_k$ to $\mu_{k+1}$ corresponding to a regularzed optimization problem (23). In this regularized objective, the original objective $f(x)$ incorporates an additional proximal term with $\lambda := 1/\eta_k$. Therefore we can analyze the convergence of $\mu_k$ by following the classical argument for gradient optimization.

*Proof of Theorem 2.* Define a filtration $\{\mathcal{H}_k\}_{k=0}^{K-1}$ with $\mathcal{H}_k$ be the information accumulated after $k$ rounds of Alg.1.

$$\mathcal{H}_0 := \sigma(\bar{\mu}),$$
$$\mathcal{H}_k := \sigma\left(\mathcal{H}_{k-1}, \sigma\left(z_{k-1,1}, \ldots, z_{k-1,B_{k-1}}\right)\right), \quad k \in [K].$$

Define the expectation of samples generated at $k$-th round as

$$\mu_k := \mathbb{E}[z_{k,i} \mid \mathcal{H}_{k-1}], \quad k \in [K-1].$$

Applying Corollary 1, we get the pre-trained score as

$$s_\theta(x_t, t) = -\left(\alpha^2(t)\bar{\Sigma} + h(t)I_D\right)^{-1}(x_t - \alpha(t)\bar{\mu}),$$

If we set $y_k$ as follows

$$y_k = \eta \cdot \left(\sigma^2 + g_k^\top \bar{\Sigma} g_k\right) + g_k^\top \bar{\mu},$$

where $\eta = 1/\lambda$. And we choose $\beta(t)$ at $k$-round as $\beta(t) = \frac{1}{2}\left(\sigma^2 + g_{k-1}^\top \bar{\Sigma}^{-1}\left(I_D + \alpha^2(t)\bar{\Sigma}/h(t)\right)^{-1} g_{k-1}\right)^{-1}$. Then, Lemma 3 provides the generated distribution in $k$-th round:

$$\mathcal{N}\left(\bar{\mu} + \eta \bar{\Sigma} g_{k-1}, \bar{\Sigma} - \frac{\bar{\Sigma} g_{k-1} g_{k-1}^\top \bar{\Sigma}}{\sigma^2 + g_{k-1}^\top \bar{\Sigma} g_{k-1}}\right). \tag{25}$$

Define the empirical covariance matrix of the latent variable $U$ as $\bar{\Sigma}_u = \mathbb{E}_{x \in \mathcal{D}, u = A^\top x}[(u - \bar{u})(u - \bar{u})^\top]$ where $\bar{u} = \mathbb{E}_{\mathcal{D}}[u]$. Then in the subspace setting, the empirical mean and covariance of data $X$ can be written as $\bar{\mu} = AA^\top \bar{\mu}$ and $\bar{\Sigma} = A\bar{\Sigma}_u A^\top$ respectively. The mean of the sample $z_{k,i}$ follows

$$\mu_k = \mathbb{E}[z_{k,i} \mid \mathcal{H}_{k-1}] = AA^\top \bar{\mu} + \eta \cdot A\bar{\Sigma}_u A^\top g_{k-1},$$

where $g_{k-1} = \nabla f(\bar{z}_{k-1})$ and $\bar{z}_{k-1} = (1/B)\sum_i^B z_{k-1,i}$. We rearrange the update rule to show a gradient ascent formula as follows

$$\mu_k = AA^\top \mu_{k-1} - AA^\top(\mu_{k-1} - \bar{\mu}) + \eta \cdot A\bar{\Sigma}_u A^\top \nabla f(\mu_{k-1}) + \eta \cdot A\bar{\Sigma}_u A^\top (g_{k-1} - \nabla f(\mu_{k-1}))$$
$$= AA^\top \mu_{k-1} - A\bar{\Sigma}_u A^\top A\bar{\Sigma}_u^{-1} A^\top (\mu_{k-1} - \bar{\mu}) + \eta \cdot A\bar{\Sigma}_u A^\top \nabla f(\mu_{k-1}) + \eta \cdot A\bar{\Sigma}_u A^\top (g_{k-1} - \nabla f(\mu_{k-1}))$$
$$= AA^\top \mu_{k-1} + \eta \cdot A\bar{\Sigma}_u A^\top \left[\nabla f(\mu_{k-1}) - \lambda A\bar{\Sigma}_u^{-1} A^\top (\mu_{k-1} - \bar{\mu})\right] + \eta \cdot A\bar{\Sigma}_u A^\top (g_{k-1} - \nabla f(\mu_{k-1})).$$

where $\lambda = 1/\eta$. Define $h(x) := f(x) - \lambda/2 \|x - \bar{\mu}\|_{\bar{\Sigma}^{-1}}^2$, we have

$$\mu_k = AA^\top \mu_{k-1} + \eta \cdot \bar{\Sigma} \nabla h(\mu_{k-1}) + \eta \cdot \bar{\Sigma}(g_{k-1} - \nabla f(\mu_{k-1})).$$

Recall the notation for the optimum: $x_{A,\lambda}^\star = \mathrm{argmax}_{x=Au} h(x)$. We consider the distance of $\mu_k$ to $x_{A,\lambda}^\star$ under the semi-norm $\|\cdot\|_{\bar{\Sigma}^{-1}}$.

$$\|\mu_k - x_{A,\lambda}^\star\|_{\bar{\Sigma}^{-1}} = \|\mu_{k-1} - x_{A,\lambda}^\star + \eta\bar{\Sigma}\nabla h(\mu_{k-1}) + \eta\bar{\Sigma}(g_{k-1} - \nabla f(\mu_{k-1}))\|_{\bar{\Sigma}^{-1}}$$
$$\leq \underbrace{\|\mu_{k-1} - x_{A,\lambda}^\star + \eta\bar{\Sigma}\nabla h(\mu_{k-1})\|_{\bar{\Sigma}^{-1}}}_{:=I_1} + \underbrace{\|\eta\bar{\Sigma}(g_{k-1} - \nabla f(\mu_{k-1}))\|_{\bar{\Sigma}^{-1}}}_{:=I_2}. \tag{26}$$

We bound the second term $I_2$ first. According to $f$ is $L$-smooth with respect to $\|\cdot\|_{\bar{\Sigma}^{-1}}$, we have

$$I_2 = \eta \|g_{k-1} - \nabla f(\mu_{k-1})\|_{\bar{\Sigma}} \leq \eta L \|\bar{z}_{k-1} - \mu_{k-1}\|_{\bar{\Sigma}^{-1}},$$

Lemma 3 shows the distribution of $z_{k-1,i}$. Therefore, according to concentration inequality for Gaussian distribution, with the probability at least $1 - \delta/K$, it holds

$$\|\bar{z}_{k-1} - \mu_{k-1}\|_{\bar{\Sigma}^{-1}}^2 \leq 2\log\left(\frac{2K}{\delta}\right) \cdot \frac{\mathrm{trace}\left(\mathbb{V}(z_{k-1,i}) \cdot \bar{\Sigma}^{-1}\right)}{B_{k-1}}.$$

We have $\mathrm{trace}\left(\mathbb{V}(z_{k-1,i}) \cdot \bar{\Sigma}^{-1}\right) \leq \mathrm{trace}\left(\bar{\Sigma} \cdot \bar{\Sigma}^{-1}\right) = d$. Therefore, $I_2$ is bounded by

$$I_2 \leq M_0/\sqrt{B_{k-1}},$$

where $M_0 := \eta L \sqrt{2 \log\left(\frac{2K}{\delta}\right) \cdot d}$. Next, we consider the first term in (26). Since $x^\star_{A,\lambda}$ is the optimum of $h$ within $\mathrm{Span}(A)$, the gradient $\nabla h(x^\star_{A,\lambda})$ is in the orthogonal subspace, i.e., $A^\top \nabla h(x^\star_{A,\lambda}) = 0$, thus $\bar{\Sigma} \nabla h(x^\star_{A,\lambda}) = 0$. The first term in (26) can be written as

$$I_1^2 = \left\| \left(\mu_{k-1} - x^\star_{A,\lambda}\right) + \eta \bar{\Sigma}\left(\nabla h(\mu_{k-1}) - \nabla h(x^\star_{A,\lambda})\right)\right\|^2_{\bar{\Sigma}^{-1}}$$
$$= \left\|\mu_{k-1} - x^\star_{A,\lambda}\right\|^2_{\bar{\Sigma}^{-1}} + \eta^2 \left\|\nabla h(\mu_{k-1}) - \nabla h(x^\star_{A,\lambda})\right\|^2_{\bar{\Sigma}}$$
$$+ 2\left\langle\mu_{k-1} - x^\star_{A,\lambda}, \eta\left(\nabla h(\mu_{k-1}) - \nabla h(x^\star_{A,\lambda})\right)\right\rangle.$$

Recall $h$ is $f$ adding a $\|\cdot\|_{\bar{\Sigma}^{-1}}$ regularized term. We get $h$ is $(L + \lambda)$-smooth with respect to semi norm $\|\cdot\|_{\bar{\Sigma}^{-1}}$ which is derived from $f$ $L$-smooth. Also, $h$ is $\lambda$-strongly concave with respect to semi norm $\|\cdot\|_{\bar{\Sigma}^{-1}}$ since $f$ is concave. According to Lemma 4, we derive

$$\left\langle\mu_{k-1} - x^\star_{A,\lambda}, \nabla h(\mu_{k-1}) - \nabla h(x^\star_{A,\lambda})\right\rangle \leq -\frac{\lambda(L+\lambda)}{L+2\lambda}\left\|\mu_{k-1} - x^\star_{A,\lambda}\right\|^2_{\bar{\Sigma}^{-1}} - \frac{1}{L+2\lambda}\left\|\nabla h(\mu_{k-1}) - \nabla h(x^\star_{A,\lambda})\right\|^2_{\bar{\Sigma}}.$$

Plugin the formula of $I_1$, we get

$$I_1^2 \leq \left(1 - \frac{2\eta\lambda(L+\lambda)}{L+2\lambda}\right)\left\|\mu_{k-1} - x^\star_{A,\lambda}\right\|^2_{\bar{\Sigma}^{-1}} + \left(\eta^2 - \frac{2\eta}{L+2\lambda}\right)\left\|\nabla h(\mu_{k-1}) - \nabla h(x^\star_{A,\lambda})\right\|^2_{\bar{\Sigma}}.$$

Since $\eta = 1/\lambda$, it holds $\eta^2 - \frac{2\eta}{L+2\lambda} > 0$. Due to $h$ $(L+\lambda)$-smoothness, we get

$$I_1^2 \leq \left(1 - \frac{2\eta\lambda(L+\lambda)}{L+2\lambda}\right)\left\|\mu_{k-1} - x^\star_{A,\lambda}\right\|^2_{\bar{\Sigma}^{-1}} + \left(\eta^2 - \frac{2\eta}{L+2\lambda}\right)(L+\lambda)^2\left\|\mu_{k-1} - x^\star_{A,\lambda}\right\|^2_{\bar{\Sigma}^{-1}}$$
$$= (1 - \eta(L+\lambda))^2\left\|\mu_{k-1} - x^\star_{A,\lambda}\right\|^2_{\bar{\Sigma}^{-1}},$$

thus, we get the bound of $I_1$

$$I_1 \leq \zeta\left\|\mu_{k-1} - x^\star_{A,\lambda}\right\|_{\bar{\Sigma}^{-1}},$$

where $\zeta := |1 - \eta(L+\lambda)|$. Combing the upper bound of $I_1$ and $I_2$, we get with probability at least $1 - \delta/K$, for $1 < k \leq K$,

$$\left\|\mu_k - x^\star_{A,\lambda}\right\|_{\bar{\Sigma}^{-1}} \leq \zeta\left\|\mu_{k-1} - x^\star_{A,\lambda}\right\|_{\bar{\Sigma}^{-1}} + \frac{M_0}{\sqrt{B_{k-1}}}.$$

As for $k = 1$, by similar derivation, we can obtain $\left\|\mu_1 - x^\star_{A,\lambda}\right\|_{\bar{\Sigma}^{-1}} \leq \zeta\left\|z_0 - x^\star_{A,\lambda}\right\|_{\bar{\Sigma}^{-1}}$. By induction, we get with probability at least $1 - ((K-1)/K)\delta$,

$$\left\|\mu_K - x^\star_{A,\lambda}\right\|_{\bar{\Sigma}^{-1}} \leq \zeta^K\left\|z_0 - x^\star_{A,\lambda}\right\|_{\bar{\Sigma}^{-1}} + M_0\sum_{k=1}^{K-1}\frac{\zeta^{K-k-1}}{\sqrt{B_k}}.$$

Choose $B_k \geq \zeta^{-4k}(1-\zeta)^{-2}$ for all $k \in [K-1]$, then we can get

$$\left\|\mu_K - x^\star_{A,\lambda}\right\|_{\bar{\Sigma}^{-1}} \leq \zeta^K\left(\left\|z_0 - x^\star_{A,\lambda}\right\|^2_{\bar{\Sigma}^{-1}} + M_1 \cdot \sqrt{d}\right) \tag{27}$$

where $M_1 := \eta L\sqrt{2\log\left(\frac{2K}{\delta}\right)}$. Since $h$ is $(L+\lambda)$-smooth with respect to $\|\cdot\|_{\bar{\Sigma}^{-1}}$, it holds

$$\left|h(\mu_K) - h(x^\star_{A,\lambda}) - \left\langle\nabla h(x^\star_{A,\lambda}), \mu_K - x^\star_{A,\lambda}\right\rangle\right| \leq \frac{L+\lambda}{2}\left\|\mu_K - x^\star_{A,\lambda}\right\|^2_{\bar{\Sigma}^{-1}}.$$

Considering that $\nabla h(x^\star_{A,\lambda}) \perp \mathrm{Span}(A)$ yields $\left\langle\nabla h(x^\star_{A,\lambda}), \mu_K - x^\star_{A,\lambda}\right\rangle = 0$, we obtain the following by rearranging the equation above

$$f\left(x^\star_{A,\lambda}\right) - f(\mu_K) \leq \frac{\lambda}{2}\left(\left\|x^\star_{A,\lambda} - \bar{\mu}\right\|^2_{\bar{\Sigma}^{-1}} - \left\|\mu_K - \bar{\mu}\right\|^2_{\bar{\Sigma}^{-1}}\right) + \frac{L+\lambda}{2}\left\|\mu_K - x^\star_{A,\lambda}\right\|^2_{\bar{\Sigma}^{-1}} \tag{28}$$
$$\leq \left[\lambda\left\|\bar{\mu} - x^\star_{A,\lambda}\right\|_{\bar{\Sigma}^{-1}}\left\|\mu_K - x^\star_{A,\lambda}\right\|_{\bar{\Sigma}^{-1}} + (L+\lambda)\left\|\mu_K - x^\star_{A,\lambda}\right\|^2_{\bar{\Sigma}^{-1}}\right].$$

Substitute (27) into above upper bound, with $z_0 = \bar{\mu}$ we have

$$f\left(x^\star_{A,\lambda}\right) - f(\mu_K) \lesssim \zeta^K \cdot (L+\lambda)\left[\left\|\bar{\mu} - x^\star_{A,\lambda}\right\|^2_{\bar{\Sigma}^{-1}} + M_1^2 d\right].$$

Since $\left\| \bar{\mu} - x_{A,\lambda}^* \right\|_{\bar{\Sigma}^{-1}}^2 = \left\| A^\top \left( \bar{\mu} - x_{A,\lambda}^* \right) \right\|_{\bar{\Sigma}_u^{-1}}^2$ is the distance within $\mathrm{Span}(A)$, i.e., $\mathcal{O}(d)$. Recall $\eta = 1/\lambda$, $\zeta = |1 - \eta(L + \lambda)| = L/\lambda$, $\lambda > L$, and $M_1 = \eta L \sqrt{2 \log\left( \frac{2K}{\delta} \right)}$. Therefore, we get the final result:

$$f\left( x_{A,\lambda}^* \right) - f(\mu_K) \lesssim \lambda \left( \frac{L}{\lambda} \right)^K d \log\left( \frac{K}{\delta} \right), \quad \text{w.p.} 1 - \delta.$$

$\square$

### E.3 Proof of Theorem 3

We first provide a proof sketch.

**Proof Sketch** The proof idea is similar to the proof of Theorem 2. For simplicity, we analyze the case where only the most recent sample batch $\mathcal{D}_k$ is merged with $\mathcal{D}_0$ for finetuning the score function. More specifically, we let $w_{k,i} = 0$ for $0 < i < k$ and $w_{k,0} = 1 - w_{k,k}$. Similar to the proof of Theorem 4, we obtain a recursive update rule given by

$$\mu_{k+1} = \bar{z}_k + \eta_k \bar{\Sigma} \left[ \nabla f(\bar{z}_k) - (1 - w_{k,k}) \eta_k^{-1} \cdot \bar{\Sigma}^{-1} (\bar{z}_k - \bar{\mu}) \right], \tag{29}$$

where $\bar{z}_k \approx \mu_k$ is the empirical mean of previous samples. This update rule also closely resembles the gradient ascent iteration for maximizing a regularized objective. A key difference here is that we can control the weights $w_{k,i}$ to reduce the impact of $\mathcal{D}_0$ and make the regularization term vanish to zero. Thus the mean $\mu_k$ eventually converges to the global maxima.

*Proof.* Define

$$\mathcal{H}_0 := \sigma(\bar{\mu}),$$
$$\mathcal{H}_k := \sigma\left( \mathcal{H}_{k-1}, \sigma\left( z_{k-1,1}, \ldots, z_{k-1,B_{k-1}} \right) \right), \quad k \in [K-1],$$
$$\mu_k := \mathbb{E}[z_{k,i} \mid \mathcal{H}_{k-1}], \quad k \in [K].$$

According to Lemma 2, with freezing $C_t$ in class (12), the pre-trained score in Round $k$ is $s_{\theta_{k+1}}(x_t, t) = - \left( \alpha^2(t) \bar{\Sigma} + h(t) I_D \right) (x_t - \alpha(t) \bar{x}_k)$ where $\bar{x}_k = \sum_{j=0}^k w_{k,j} \bar{z}_j$ and $\bar{z}_j = \mathbb{E}_{x \in \mathcal{D}_j}[x]$. By choosing $y_k$, and weights $w_{k,j}$ as

$$y_k = \eta_k \cdot \left( \sigma^2 + g_k^\top A \bar{\Sigma}_u A^\top g_k \right) + g_k^\top A A^\top \bar{x}_k,$$
$$w_{k,0} = 1 - w_k$$
$$w_{k,j} = 0, \; 1 \le j < k,$$
$$w_{k,k} = w_k,$$

where $\eta_k > 0$, $0 < w_k < 1$ will be specified later. And we choose $\beta(t)$ at Round $k$ as $\beta(t) = \frac{1}{2} \left( \sigma^2 + g_{k-1}^\top \bar{\Sigma}^{-1} \left( I_D + \alpha^2(t) \bar{\Sigma}/h(t) \right)^{-1} g_{k-1} \right)^{-1}$. Lemma 3 gives the mean of distribution of $z_{k+1,i}$ as

$$\mu_{k+1} = \bar{x}_k + \eta_k \bar{\Sigma} g_k, \tag{30}$$

and the output distribution

$$\mathcal{N}\left( \bar{x}_{K-1} + \eta_{k-1} \bar{\Sigma} g_{K-1}, \bar{\Sigma} - \frac{\bar{\Sigma} g_{K-1} g_{K-1}^\top \bar{\Sigma}}{\sigma^2 + g_{K-1}^\top \bar{\Sigma} g_{K-1}} \right). \tag{31}$$

Applying Lemma 3 yields $z_{k,i} \in \mathrm{Span}(A)$, thus, $\bar{x}_k = AA^\top \bar{x}_k$ and $\bar{\Sigma} = A \bar{\Sigma}_u A^\top$, we get the update rule reduced to

$$\mu_{k+1} = AA^\top \left( (1 - w_k) \bar{\mu} + w_k \bar{z}_k \right) + \eta_{k-1} A \bar{\Sigma}_u A^\top g_{k-1}$$
$$= AA^\top \bar{z}_k + \eta_k A \bar{\Sigma}_u A^\top \left( \nabla f(\bar{z}_k) - \eta_k^{-1}(1 - w_k) A \bar{\Sigma}_u^{-1} A^\top (\bar{z}_k - \bar{\mu}) \right).$$

We set $w_k = 1 - \eta_k \lambda$ and set $\eta_k = \eta$, where $\lambda, \eta > 0$ will be specified later. Therefore, we have

$$\mu_{k+1} = AA^\top \bar{z}_k + \eta A \bar{\Sigma}_u A^\top \nabla h_\lambda(\bar{z}_k), \tag{32}$$

where $h_\lambda(x) := f(x) - (\lambda/2) \|x - \bar{\mu}\|^2_{\bar{\Sigma}^{-1}}$. Define $x^\star_{A,\lambda} = \text{argmax}_{x = Au} h_\lambda(x)$. With some similar steps in proof in Appendix E.2, by choosing $B_k \geq \zeta^{-4k}(1 - \zeta)^{-2}$, together with $z_0 = \bar{\mu}$, we get

$$\|\mu_K - x^\star_{A,\lambda}\|_{\bar{\Sigma}^{-1}} \lesssim \zeta^K \left( \|\bar{\mu} - x^\star_{A,\lambda}\|_{\bar{\Sigma}^{-1}} + M_1 \cdot \sqrt{d} \right), \quad \text{w.p. } 1 - \delta,$$

with $\eta = \dfrac{2}{L + 2\lambda}$, $\zeta = |1 - \eta(L + \lambda)|$ and $M_1 = 2L\sqrt{(1 + \eta^2) \log \left( \dfrac{2K}{\delta} \right)}$. Also, we can get (28) as in proof in Appendix E.2. We restate it here:

$$f\left(x^*_{A,\lambda}\right) - f(\tilde{z}_K) \leq \frac{\lambda}{2} \left( \|x^*_{A,\lambda} - \bar{\mu}\|^2_{\bar{\Sigma}^{-1}} - \|\mu_K - \bar{\mu}\|^2_{\bar{\Sigma}^{-1}} \right) + \frac{L + \lambda}{2} \|\mu_K - x^*_{A,\lambda}\|^2_{\bar{\Sigma}^{-1}}. \quad (33)$$

Since $f$ is concave,

$$f\left(x^*_A\right) - f\left(x^*_{A,\lambda}\right) \leq \left\langle \nabla f\left(x^*_{A,\lambda}\right), x^*_A - x^*_{A,\lambda} \right\rangle = \lambda \left\langle \bar{\Sigma}^{-1}(x^*_{A,\lambda} - \bar{\mu}), x^*_A - x^*_{A,\lambda} \right\rangle.$$

Adding (33), it holds

$$f\left(x^*_{A,\lambda}\right) - f(\mu_K) \leq \frac{\lambda}{2} \left( \|x^*_{A,\lambda} - \bar{\mu}\|^2_{\bar{\Sigma}^{-1}} - \|x^*_{A,\lambda} - x^*_A\|^2_{\bar{\Sigma}^{-1}} - \|\mu_K - \bar{\mu}\|^2_{\bar{\Sigma}^{-1}} \right) + \frac{L + \lambda}{2} \|\mu_K - x^*_{A,\lambda}\|^2_{\bar{\Sigma}^{-1}}.$$

Due to (32), we have, it holds w.p. $1 - \delta$,

$$f\left(x^*_{A,\lambda}\right) - f(\mu_K) \lesssim \left[ \lambda \|x^*_A - \bar{\mu}\|^2_{\bar{\Sigma}^{-1}} + (L + \lambda)\zeta^K \left( \|\bar{\mu} - x^*_{A,\lambda}\|^2_{\bar{\Sigma}^{-1}} + M_1 d \right) \right].$$

We choose $\lambda = L \log K / (4K)$ and get

$$f\left(x^*_{A,\lambda}\right) - f(\mu_K) \lesssim \frac{L \log K}{K} \cdot \left[ \|x^*_A - \bar{\mu}\|^2_{\bar{\Sigma}^{-1}} + \|\bar{\mu} - x^*_{A,\lambda}\|^2_{\bar{\Sigma}^{-1}} + M_1 d \right], \quad \text{w.p. } 1 - \delta.$$

With assuming $\left\| x^\star_{A,\lambda} \right\|$ is bounded, we derive

$$f\left(x^*_{A,\lambda}\right) - f(\mu_K) = \mathcal{O}\left( \frac{dL^2 \log K}{K} \cdot \log\left( \frac{K}{\delta} \right) \right), \quad \text{w.p. } 1 - \delta.$$

### E.4 Auxiliary Lemma

The following is a standard result in convex optimization utilized in previous proofs.

**Lemma 4.** *Let $f$ be $\alpha$-strongly concave and $\beta$-smooth with respect to the (semi) norm $\|\cdot\|_{\Sigma^{-1}}$, for all $x$ and $y$, it holds*

$$-\langle \nabla f(x) - \nabla f(y), x - y \rangle \geq \frac{\alpha\beta}{\alpha + \beta} \|x - y\|^2_{\Sigma^{-1}} + \frac{1}{\alpha + \beta} \|\nabla f(x) - \nabla f(y)\|^2_{\Sigma}. \quad (34)$$

*Proof.* See Bubeck et al. [9, Lemma 3.11] for a proof. $\square$

$\square$

## F Additional Materials for Experiments

### F.1 Simulation

We experiment with our design of the gradient guidance as well as Algorithm 1 and Algorithm 2. Going beyond our theoretical assumptions, we adopt a 15M-parameter U-Net as the score function class for training and fine-tuning our diffusion model.

#### F.1.1 Experiment Setup

For linear data structure, we set the data's ambient dimension as $D = 64$ and the linear subspace dimension as $d = 16$. The linear subspace is represented by an orthogonal matrix $A \in \mathbb{R}^{D \times d}$. We randomly generate a matrix $A$ and fix it once generated. After that, we sample a data point $X$ by first randomly sampling a latent variable $U \sim \mathcal{N}(0, I_d)$ and computing $X = AU$. We independently sample a total of 65536 data points as our pre-training data set. For nonlinear data structure, data are uniformly sampled from a unit ball in $\mathbb{R}^{64}$.

The objective functions considered in our experiments are $f_1(x) = 10 - (\theta^\top x - 3)^2$ and $f_2(x) = 5 - 0.5\|x - b\|$. Here, $\theta$ and $b$ are randomly generated and fixed afterward. Since our data assumes a low-dimensional subspace representation, it is convenient to decompose $\theta$ into $\theta_\perp = (I - AA^\top)\theta$

and $\theta_{\parallel} = AA^{\top}\theta$, representing the off-support and on-support components. We refer to $\frac{\|\theta_{\perp}\|}{\|\theta_{\parallel}\|}$ as the off/on-support ratio. Analogously, for a generated sample, we can also define its off/on-support ratio. Clearly, a small off/on-support ratio indicates close vicinity to the subspace.

**Score Network Pre-training.** We utilize a version of the U-Net [54], with 14.8M trainable parameters. Note that this is a complicated network going beyond the linear score function class considered in our theories. Following the implementation of Denoising Diffusion Probabilistic Models (DDPM, Ho et al. [34]), we train the U-Net o estimate the score function $\nabla \log p_t$, via minimizing the score matching loss introduced in Eqn. (3). We discretize the backward process to have 200 time steps as in Nichol and Dhariwal [47], and the U-Net is trained using our generated data set for 20 epochs. We use Adam as the optimizer, set the batch size as 32, and set the learning rate to be $10^{-4}$. After the pre-training phase, we confirmed that the data subspace structure is well learned, as the generated samples using the pre-trained diffusion model have an average off/on-support ratio of 0.039.

**Implementation of Algorithm 1.** In each iteration of Algorithm 1, we need to compute the gradient guidance $\mathtt{G}_{loss}$. We set the targeted $y$ value at the $k$-th iteration as $y_k = \delta + g_k^{\top} z_k$, where $\delta$ specifies the increment per iteration. The choice on $\delta$ is instance-dependent and we set it via tuning for near-optimal in different experiments. For comparing naive gradient with gradient guidance in Figure 8, we set $\delta = 0.2$ and $0.9$, respectively for using naive gradient $\mathtt{G}$ and gradient guidance $\mathtt{G}_{loss}$. In Figure 9, we choose $\delta$ to be (a) 0.05, (b) 0.2, (c) 1, and (d) 1, corresponding to each panel. We initialize Algorithm 1 with a batch of 32 samples generated by the pre-trained model. Each sample determines an optimization trajectory. We repeat Algorithm 1 for 5 times with different random seeds and report the error bars.

**Implementation of Algorithm 2.** Algorithm 2 differs from Algorithm 1 in that it allows additional fine-tuning of the pre-trained score network. In practice, to update the score network incorporating newly generated data, one does not have to exactly solve (14) by re-training the full model from scratch. Instead, (14) can be viewed as a guideline that motivates more computationally efficient ways for updating the pre-trained score. It is a common practice to only **fine-tune** the weights of the old model by performing gradient descent over a few batches of newly generated data, which is similar to the spirit of (14). To be more specific, we adopt a computationally lightweight fine-tuning strategy: We only perform one Adam optimization step using the re-weighted loss given by Eqn. (14) with a batch of 32 generated samples. We set the learning rate as $10^{-6}$. This simple strategy already demonstrates good performances as shown in Figure 10. Other implementation details are kept the same as those of Algorithm 1.

### F.1.2   Results

We first demonstrate our gradient guidance $\mathtt{G}_{loss}$ preserves the subspace structure learned from the pre-trained model. For comparison, we also tested the naive guidance $\mathtt{G}$ defined following Proposition 3 (with $\Sigma = I$). For a quick reference, we repeat the definition here:

$$\mathtt{G}(x_t, t) := \beta(t) \left( y - g^{\top} \mathbb{E}[x_0|x_t] \right) g,$$

where $\beta(t) > 0$ and $y \in \mathbb{R}$ are tuning parameters, and $\mathbb{E}[x_0|x_t]$ is the conditional expectation of $x_0$ given noise corrupted data $x_t$. For implementation, we replace $\mathbb{E}[x_0|x_t]$ by its look-ahead estimator $\hat{\mathbb{E}}[x_0|x_t]$ based on the Tweedie's formular.

**Comparing $\mathtt{G}$ and $\mathtt{G}_{loss}$ on Preserving Subspace Structure.** Figure 8 (a), (c) verify that the naive gradient $\mathtt{G}$ performs much worse than $\mathtt{G}_{loss}$ in preserving the linear subspace structure. It is consistent with our theoretical finding that the gradient guidance $\mathtt{G}_{loss}$ keeps the generated sample close to the latent subspace, with substantially smaller off-support errors. When allowing adaptive score fine-tuning in Algorithm 2, Figure 8 (b), (d) show that the off-support error increases as the model gets fine-tuned using self-generated data, due to increasing distribution shift. Even in this case, the naive gradient $\mathtt{G}$ leads to much more severe off-support errors as compared to $\mathtt{G}_{loss}$.

**Algorithm 1 Converges to Regularized Optima.** We plot the convergence of Algorithm 1 in terms of the objective value in Figure 9. Figure 9 (a),(b) are for the objective function $f_1 = 10 - (\theta^{\top}x - 3)^2$ as the objective function, while Figure 9(c),(d) are for the objective $f_2 = 5 - 0.5\|x - b\|$. We observe

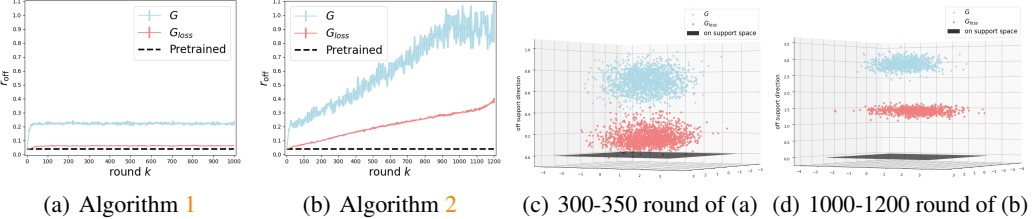

(a) Algorithm 1       (b) Algorithm 2       (c) 300-350 round of (a)    (d) 1000-1200 round of (b)

Figure 8: **Comparison between two types of gradient guidance** G and G$_{loss}$. We plot the off/on support ratio of the generated samples, denoted by $r_{\text{off}} = \frac{\|x_\perp\|}{\|x_\parallel\|}$. The objective function is $f_1(x)$, with $\theta$ having an off/on-support ratio of 9.

that the algorithm converges to reach some sub-optimal objective value, but there remains a gap to the maximal value. This is consistent with our theory that the pre-trained model essentially acts as a regularization in addition to the objective function. Adding gradient guidance alone cannot reach global maxima. This coincides with our theoretical findings in Theorem 2.

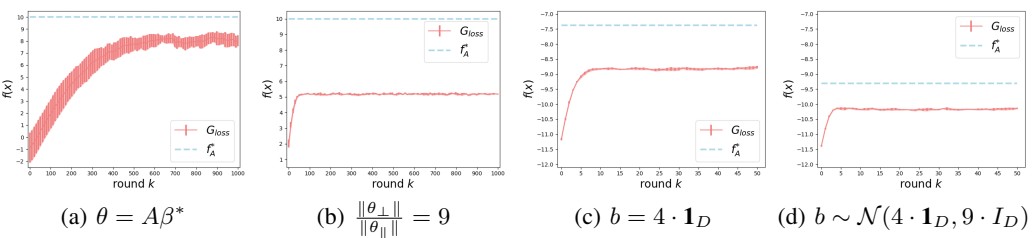

(a) $\theta = A\beta^*$     (b) $\frac{\|\theta_\perp\|}{\|\theta_\parallel\|} = 9$     (c) $b = 4 \cdot \mathbf{1}_D$     (d) $b \sim \mathcal{N}(4 \cdot \mathbf{1}_D, 9 \cdot I_D)$

Figure 9: **Convergence of Algorithm 1 under different objectives**. Objectives are $f_1(x)$ for (a) and (b), and $f_2(x)$ for (c) and (d). Parameters $\theta$ and $b$ are specified as (a) $\theta = A\beta^*$ with $\beta^*$ being sampled from the unit ball in $\mathbb{R}^d$; (b) the off/on-support ratio of $\theta$ being 9 (same as Figure 8); (c) and (d) choosing $b$ as a homogeneous vector or randomly from a Gaussian distribution. All the experiments adopt the gradient guidance G$_{loss}$.

**Algorithm 2 Converges to Global Optima.** Algorithm 2 converges to the maximal value of the objective function $f_1 = 10 - (\theta^\top x - 3)^2$ as shown in Figure 10(a). In Figure 10(b), we visualize the distribution of generated samples of Algorithm 1 (blue) and 2 (red), respectively, as the iteration evolves. We see that samples from Algorithm 1 mostly stay close to the pre-training data distribution (area described by the dotted contour). In constrast, samples of Algorithm 2 move outside the contour, as the diffusion model gets fine-tuned using self-generated data.

**Results for Nonlinear Data Structure.** We apply Algorithm 1 to data uniformly sampled from a unit ball in $\mathbb{R}^{64}$. The objective reward function is defined as $f(x) = \theta^\top x$, where $\|\theta\| = 1$. The left panel of Figure 11 demonstrates that rewards increase and converge when using Algorithm 1. Higher guidance strength $\Delta$ (corresponding to lower regularization) results in a higher convergent reward. The right panel of Figure 11 shows that, for the same reward level, gradient guidance achieves a smaller deviation from the unit ball compared to the naive gradient approach. This suggests that gradient guidance can better preserve data structure for nonlinear manifolds.

### F.2 Image Generation

**Hyperparameters.** Since tuning parameters $y$ and $\beta(t)$ are both for the strength of guidance, we can fix one of them. We choose $\beta(t) = 100$ as suggested by [71], and set a series of $y \in \{2, 4, 10\}$. We run batch size $= 20$ samples parallelly across iterations to evaluate the rewards. The prompt is uniformly sampled from the 1000 classes of ImageNet [21].

### F.3 Time Efficiency

We summarize the time cost of our experiments on one NVIDIA A100 GPU in Table 1.

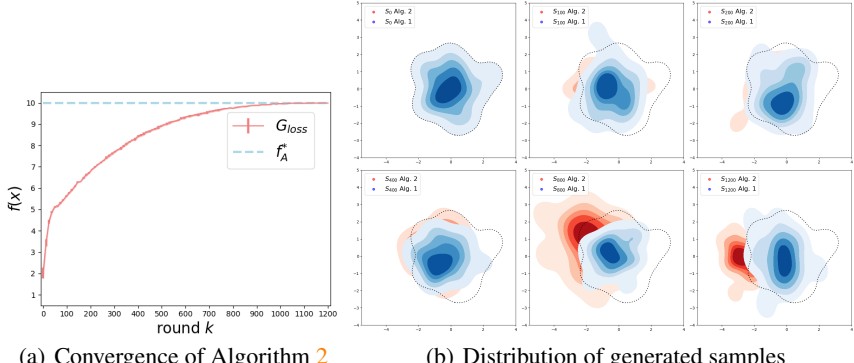

(a) Convergence of Algorithm 2          (b) Distribution of generated samples

Figure 10: **Convergence of Algorithm 2**. Panel (a) plots the objective values achieved by Algorithm 2 as a function of iterations. Here $\theta$ is chosen the same as in Figure 9 (b) with off/on-support ratio $\frac{\|\theta_\perp\|}{\|\theta_\|\|} = 9$. Panel (b) visualizes the distribution of the generated samples of Algorithm 2 (red) across the iterations. For comparison, we also visualize the distribution of generated samples of Algorithm 1 (blue).

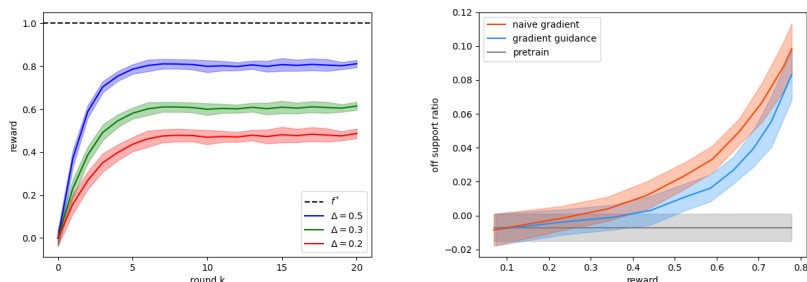

Figure 11: **Nonlinear data structure experiment.** We apply Algorithm 1 to data uniformly sampled from a unit ball in $\mathbb{R}^{64}$. The objective reward function is $f(x) = \theta^\top x$, where $\|\theta\| = 1$. **Left:** Rewards increase and converge with Algorithm 1. Higher guidance strength $\Delta$ (lower regularization) results in a higher convergent reward. **Right:** For the same reward, gradient guidance achieves a smaller deviation from the unit ball compared to the naive gradient. This indicates that gradient guidance can preserve data structure for nonlinear manifolds.

| | Total runtime (iterations) | | Per iteration | No guidance |
|---|---|---|---|---|
| Simulation | 3.8 min (50 iter) | 76 min (1000 iter) | 4.6 s | 2.6 s |
| Image | | 1.3 min (5 iter) | 15.8 s | 4.9 s |

Table 1: Runtime Efficiency of Algorithm 1. Red refers to the total time to converge. No guidance refers to the time for one-time inference of the pre-trained model.

