# OpenReview forum: "Gradient Guidance for Diffusion Models: An Optimization Perspective"
_NeurIPS.cc/2024/Conference — NeurIPS 2024 poster_

### Official Review · Reviewer_HWVj · 2024-07-08

**Soundness:** 4
**Presentation:** 4
**Contribution:** 4
**Rating:** 7
**Confidence:** 4

**Summary:**

This paper investigate gradient guidance for adapting or fine-tuning pre-trained diffusion models from an optimization perspective.
The author proposed a look-ahead loss based gradient guidance and two variants of diffusion-based generative optimization algorithms utilizing it.
The author provided theoretical guarantees for adapting/fine-tuning diffusion models.

**Strengths:**

Overall, although this is a theory paper, it is very impressive. It used theoretical approaches to explain some problems of diffusion with guidance:
1. naive guidance doesn't work on latent structure of data
2.  algorithm on guided diffusion using gradient queries on new samples
3. proposed an adaptive gradient guided diffusion, where both pre-trained score network and guidance are iteratively updated using self-generated samples.

**Weaknesses:**

No obvious weakness.
If possible, please include the reason for each step in your proof, i.e. the first step follows from???, the second step follows from???. Then it will be a perfect paper. I will increase the score if you can fix this issue.

**Questions:**

No questions

---

> ### Author Rebuttal · Authors · 2024-08-07
>
> We appreciate your recognition of our theoretical contributions and your valuable suggestions! We've added more detailed explanations of the derivations in the proof.

---

### Official Review · Reviewer_uzJ1 · 2024-07-10

**Soundness:** 2
**Presentation:** 2
**Contribution:** 2
**Rating:** 4
**Confidence:** 4

**Summary:**

This paper proposes a new approach to the problem of gradient-guided generation for diffusion models. The main challenge of gradient-guided generation is to maintain the generated sample within the support of the sample distribution. To address this issue, this work starts with a simplified model, using a linear objective function and a low-dimensional linear sample space, and provides a closed form for the gradient guidance. Under this simplified model, the paper offers a theoretical analysis to show the converged distribution of the generated samples for concave and smooth objective functions. Lastly, it also proposes another variant that allows fine-tuning over pretrained models, further improving optimization performance

**Strengths:**

1. The idea of deriving this form of gradient guidance is interesting and novel, particularly the use of a low-dimensional linear sample space to demonstrate the problem of out-of-distribution in gradient-guidance sampling.

2. The theoretical results are solid, and I appreciate the finding that the proposed method can indeed preserve the low-dimensional linear structure theoretically.

**Weaknesses:**

1. Although I appreciate the motivation and analysis in this work, I do not think this method is practical in reality. Firstly, the core advantage of gradient-guidance approaches like [1][2] is that they do not require gradient backpropagation over the neural network, thus adding only minor computational cost to direct sampling. In contrast, the method proposed in this work requires gradient backpropagation over the neural network, making it prohibitively slow. As mentioned in the paragraph above Algorithm 1, guiding stable diffusion requires 76 minutes for optimization. Secondly, since the method requires gradient backpropagation over the neural network, I think the author should compare it to the serial works on direct optimization in [3][4][5], which are completely ignored in this work. These works target the same task and have the same computational requirements. From my understanding, the algorithm proposed in this work cannot rival direct optimization methods, as the latter do not rely on simplified models to derive the algorithm. In fact, a concurrent work [6] shows that one can optimize SDXL, a 3B parameter model, within 10 minutes, which is in sharp contrast to the performance of Algorithm 1 proposed in this work.

2. The theoretical analysis is certainly novel, but it heavily relies on the linear sample space assumptions. It is unclear whether this method can be applied to cases where the sample space is highly nonlinear. The author conducted a simple experiment with image diffusion, which somewhat investigates this point, but I think the author should also conduct experiments on a highly nonlinear synthetic data distribution to validate if their method works in this scenario.

3. The writing and presentation of the paper can be improved. For example, when moving from Section 3 to Section 4, the target objective function switches from a linear function to a general function without much explanation. I can only guess from Algorithm 1 that the gradient
$g$ is directly replaced by some stochastic gradient estimator of a general function $f$? I think the author should provide some explanation on this point.

[1] Song, Jiaming, et al. "Loss-guided diffusion models for plug-and-play controllable generation." International Conference on Machine Learning. PMLR, 2023.

[2] Chung, Hyungjin, et al. "Diffusion posterior sampling for general noisy inverse problems." arXiv preprint arXiv:2209.14687 (2022).

[3] Bram Wallace, Akash Gokul, Stefano Ermon, and Nikhil Naik. End-to-end diffusion latent optimization improves classifier guidance. In Proceedings of the IEEE/CVF International Conference on
Computer Vision, pages 7280–7290, 2023b.

[4] Heli Ben-Hamu, Omri Puny, Itai Gat, Brian Karrer, Uriel Singer, and Yaron Lipman. D-flow:
Differentiating through flows for controlled generation. arXiv preprint arXiv:2402.14017, 2024.

[5] Korrawe Karunratanakul, Konpat Preechakul, Emre Aksan, Thabo Beeler, Supasorn Suwajanakorn,
and Siyu Tang. Optimizing diffusion noise can serve as universal motion priors. arXiv preprint
arXiv:2312.11994, 2023.

[6] Tang, Zhiwei, et al. "Tuning-Free Alignment of Diffusion Models with Direct Noise Optimization." arXiv preprint arXiv:2405.18881 (2024).

**Questions:**

1. I wonder how the algorithm proposed in this work compares to the direct optimization approaches in [3][4][5][6] (references mentioned above)?

2. Could the author conduct some simple experiments on synthetic data with a nonlinear sample space? I am curious how well the proposed method will perform when the linear assumption is violated.

**Limitations:**

See my comments above. In summary, the main limitations of this work are:

1. Incomplete literature review and comparison with existing works.

2. The performance of the proposed algorithm appears to be impractical.

3. The presentation can be improved.

---

> ### Author Rebuttal · Authors · 2024-08-07
>
> >**Q1:** The proposed method is prohibitively slow due to gradient backpropagation over the neural network. Other approaches like [1][2] do not require gradient backpropagation over the neural network.
>
> **A1:** There seems to be some confusion. Your understanding is incorrect. Both [1] and [2] require backpropagation through the score network. Please refer to:
>
> - In [1], Eqn.2 includes score network, and Eqn.8 takes gradient over it.
>
> - In [2], Eqn.10 includes score network, and Eqn.16 takes gradient over it.
>
> Our algorithm is as efficient as other gradient-based guidance methods while enjoying additional theoretical guarantees. Please see our **new Table 1 in [pdf][pdf-link]** for a detailed analysis of our runtime efficiencies.
>
> >**Q2:** The method can't rival direct optimization. These works target the same task and have the same computational requirements.
>
> **A2:** Thank you for pointing us to these papers. They are relevant and we have added discussions about these works in our **expanded literature review** in Sec 4 of [pdf][pdf-link].
>
> - **We target more general tasks and focus on theory.** We respectfully disagree that our method and direct optimization target the same tasks. Indeed they are relevant. However, we consider general optimization over data's latent manifold. Our Section 5 provides the first methodology and theory for adapting diffusion models with self-play and online data collection to **find the global optimum of a user-provided objective function**.
>
> - **They don't have the same computation requirement: Direct optimization has a high memory burden and ours does not.**  While our gradient guidance method enables sampling using $O(1)$ memory, direct optimization methods [3-6] suffer from **O(T)** memory, a substantially higher burden, because they need to backpropagate through the ODE solver, which requires storing all intermediate gradients when utilizing the chain rule. For example, [4] states: *"backpropagating through the solver can be expensive in memory"*, similarly, [5] notes: *"maintaining the intermediate activations for solving the ODE during backpropagation can be memory-intensive. This issue can be addressed with gradient checkpointing or an invertible ODE, at the cost of more computation or model complexity."* Thus, our method is much more light-weighted.
>
> >**Q3:** A concurrent work DNO [6] shows that one can optimize SDXL, a 3B parameter model, within 10 minutes, which is in sharp contrast to the performance of Algorithm 1 proposed in this work.
>
> **A3:** Thank you for pointing out this paper. After carefully checking their paper, precise running time was only reported for SD1.5 (Table 1 in DNO[6]), the same model tested in our experiments. For SDXL, there's only a figure with a vague runtime axis, prevents direct comparision.
> We added a **new Table 1 [our rebuttal][re-link] for analysis of computation efficiency**. For adapting the SD 1.5 model, our method takes 15.8 seconds per optimization round, and <5 iterations to converge (as shown in Figure 6). **Our total runtime is < 2min**, which is comparable to the 10min runtime reported by Table 1 in DNO .
>
> >**Q4:** Could you conduct some simple experiments on synthetic data with a nonlinear sample space?
>
> **A4:** Thanks for the suggestion. We have conducted **additional experiment (Sec 1 in [pdf][pdf-link] )** on data on nonlinear space. Our new result demonstrates that our guidance better preserves the structure of nonlinear manifolds than naive gradient guidance.
>
> >**Q5:** When moving from Section 3 to Section 4, why does the target objective function switch from a linear function to a general function?
>
> **A5:** Yes our task is to generate solutions for general nonlinear, concave optimization problems. Our Section 3 uses the linear model as a motivating example to derive the gradient guidance. Our Sections 4, 5 provide our main results for general nonlinear optimization.
>
> References:
>
> [1] Song, Jiaming, et al. "Loss-guided diffusion models for plug-and-play controllable generation." International Conference on Machine Learning. PMLR, 2023.
>
> [2] Chung, Hyungjin, et al. "Diffusion posterior sampling for general noisy inverse problems." 2022.
>
> [3] End-to-end diffusion latent optimization improves classifier guidance. ICCV, 2023.
>
> [4] D-flow: Differentiating through flows for controlled generation, 2024.
>
> [5] Optimizing diffusion noise can serve as universal motion priors, 2023.
>
> [6] Tuning-Free Alignment of Diffusion Models with Direct Noise Optimization. arXiv preprint, 2024.
>
>
> [re-link]: https://openreview.net/forum?id=X1QeUYBXke&noteId=WsELdf2puF
> [pdf-link]: https://openreview.net/attachment?id=WsELdf2puF&name=pdf

---

> > ### Comment · Reviewer_uzJ1 · 2024-08-12
> > **Thanks for your rebuttal**
> >
> > I would like to thank the author for addressing some of my concerns and correcting my mistake. Thus I decided to increase my score by 1. However, I still do not lean towards acceptance for two reasons.
> >
> > 1. The design of the algorithm and theory, which are claimed to be the main contribution of this work relies on a simplified model (linear or concave). This makes it unclear what the fundamental factor to drive the improvement over other baselines like LGD, and also dims the meaningfulness of the theoretical results. Besides, I think several aspects of algorithm design also appear in the works on gradient guidance.
> >
> > 2. I disagree with the author's comment "We respectfully disagree that our method and direct optimization target the same tasks. Indeed they are relevant. However, we consider general optimization over data's latent manifold. Our Section 5 provides the first methodology and theory for adapting diffusion models with self-play and online data collection to find the global optimum of a user-provided objective function." Direct optimization method also targets the scenario "adapting diffusion models with self-play and online data collection to find the global optimum of a user-provided objective function.", and hence this work is not "the first methodology". As far as I can tell, all the experiments in the current manuscript can also be tackled by the Direct optimization method. While saying so, I do agree that the gradient guidance method has lower time complexity. I know it could hard to run the direct numerical comparison during the short rebuttal period. But I do wish the author to acknowledge the importance of this comparison because it is still possible that: The direct optimization method, while having high time-complexity for each gradient step, can still have a better reward-time trade-off compared to the gradient-guidance method.

---

> ### Author Response · Authors · 2024-08-14
>
> Hello thanks for the response! We are glad that **your primary concern about backpropagation is now resolved**. In regards to further comments:
>
> > "the main contribution of this work relies on a simplified model (linear or concave). "
>
> Sorry there could still be a major misunderstanding. Our contribution does not **rely** on a simplified model.
>
> 1. The simplified model is only used to **motivate** the design of gradient guidance. We choose to use a simple linear model to intuitively explain why certain forms of guidance are better than others.
> But our results are not limited to the simplified model, as follows.
>
> 2. **Theorem 1** holds for  **arbitrary** distributions and does **not** require a linear score model.
>
> 3. The guidance term constructed in our paper, G_loss (Eqn 7 and Eqn 11 in submission), applies to any pre-trained score network. It is not limited to linear models.
>
> 4. Our first experiment (Sec 6.1 of submission) uses a 15M-parameter, nonlinear, U-Net score network.
>
> 5. Our second experiment (Sec 6.2) generated human-interpretable images, validating that our method preserves the latent manifold structure underlying image data.
>
> 6. The image experiment (Sec 6.2) also validated the method on nonconcave objectives.
>
> 7. Per your request for additional nonlinear experiments, we have provided a new set of experiments for synthetic data on nonlinear, spherical manifolds (**Sec 1 of [pdf][pdf-link]**).
>
> While our methodology extends beyond "simplified models", we intentionally begin with simple mathematical models as the foundation for exploring complex theories. Simple models, such as linear models, have been fundamental to the developments of deep learning and diffusion model theories, e.g., [38] Marion et al, 2024. Philosophically, mastering these basic models often provides the deepest insights, most practical utility, as well as generalizability. Therefore, we firmly believe that robust theoretical research must start with simplicity.
>
>
>
> > “several aspects of algorithm design also appear in the works on gradient guidance”
>
> Our paper focuses on theory. As we already explained, gradient guidance via backpropagation is a common practice (universal guidance does it; [1],[2] also does it). Our contribution is to provide theoretical motivation for this design and establish the first optimization convergence theory.
>
> > “Direct optimization method also targets the scenario "adapting diffusion models with self-play and online data collection to find the global optimum of a user-provided objective function.", and hence this work is not "the first methodology".
>
> By “the first methodology”, we mean the first methodology to provably solve optimization problems with convergence/optimality guarantees, with both theory and experiments. We are not aware of any other method that probably adapts a pretrained diffusion model to solve optimization problems to global optimum.
>
> We appreciate that you now agree that guidance methods and direct optimization “have different computation requirements”. They are different approaches in nature, and fully understanding their limits would be future work. While our focus is guidance, we appreciate that you highlight this alternative technological route. We’d be happy to incorporate more discussions about it in our related work section.
>
> **Could you please point us to the specific literature on direct optimization with self-play and online data collection to find global optimum of any objective function?** It’s possible that we have missed something. We are happy to go over them and include them in our literature review. Thank you!
>
>
> - Please also note that the discussion period will end soon today. So we might not be able to respond again, but we appreciate any constructive comment!
>
>
>
> [1] Song, Jiaming, et al. "Loss-guided diffusion models for plug-and-play controllable generation." International Conference on Machine Learning. PMLR, 2023.
>
> [2] Chung, Hyungjin, et al. "Diffusion posterior sampling for general noisy inverse problems." arXiv preprint arXiv:2209.14687 (2022).
>
>
> [38] Marion et al. Implicit diffusion: Efficient optimization through stochastic sampling, 2024
>
>
> [pdf-link]: https://openreview.net/attachment?id=WsELdf2puF&name=pdf

---

### Official Review · Reviewer_2yVt · 2024-07-11

**Soundness:** 2
**Presentation:** 3
**Contribution:** 2
**Rating:** 6
**Confidence:** 4

**Summary:**

Under the assumption of the data belonging to a low-dimensional linear subspace, the authors investigate two common gradient-based guidance techniques of diffusion models, encouraging the use of one of them (computing the gradient at the estimate of x0 given xt, as done in many works).
For concave reward, and under the additional assumption of linear score function, they also establish convergence results for the mean of the guided process.

**Strengths:**

Overall providing theory for guided diffusion is appreciated.

The paper formally shows an advantage of a certain guidance over the other.

The presentation is clear and the paper is convenient to read.

**Weaknesses:**

The assumptions made in the paper are strong (as discussed below).

A certain weakness of Theorems 2 and 3 is that they consider only convergence of the mean of the procedure do not say anything about the variance.

There are no new algorithmic ideas in the paper, as the promoted gradient-based guidance is known and common, and also there exist works on fine-tuned/adaptive diffusion models. Thus, the authors needs to tone down some claims in the contribution list and in the abstract.

More comments are stated below.

**Questions:**

1.

The tight connection between gradient-guided diffusion models and proximal optimization algorithms is discussed in detail in:
Tomer Garber and Tom Tirer, "Image Restoration by Denoising Diffusion Models with Iteratively Preconditioned Guidance," CVPR, 2024.
This relevant literature should be mentioned.

2.

The statement in line 48:
"Why does naively guiding diffusion models using gradient never work in practice?"
Is not clear, as there are leading approaches with gradient-based guidance that work, such as (Garber and Tirer, 2024) and the back-projections [53] / least-squares [14] methods that it generalizes.
Only much later in the paper, the reader can see that the authors do not refer to **the common practice** of computing the gradient at the estimate of x0 given xt as a plain gradient-based guidance. This should be stated clearly already at early stage.
This also means that you need to tone down the claim in the contribution list that you: "introduce guidance based on forward prediction loss", because this is already a well-known practice.

3.

Based on your Eq. 1, it seems that you consider the variance preserving (VP) SDE formulation.
Why do you state in line 85 that without loss of generality all your analysis is done for q=1?
This needs to be explained, at least in the appendix, as q=1 is not the common setting (but rather some linearly increase from near 0 to 1).

4.

There seems to be some discrepancy between your Eq. 2, associated with Eq. 1, and the SDE formulation of diffusion models [50]. Recall that dt in Eq. 2 is negative.

5.

The "general" explanation in Section 3.2 that naive gradient-based guidance doesn't work is not convincing because it ignores that fact that the step-size decreases as the t get closer to t=0 and that the noise injection can mitigate error propagation.

6.

You state: "Alg. 1 is light-weighted ... takes 76min overall." Recall that the reverse process is not performed in the offline pretraining stage. So obviously Alg. 1 is quite slow. Indeed, it is known that guidance methods that include the DNN's Jacobian computation are slow.

7.

The assumptions of the theoretical analyses in the paper are quite strong.
Especially Assumption 1 on the signal belonging to low dimensional linear subspace and the assumption on linear score model (Eq. 12).
Already in the contribution list you should state the assumption on the score model, which essentially leads to convex optimization.

8.

Any idea how can Theorem 1 be generalized to low dimensional manifold rather than low dimensional linear subspace?
Under the assumptions, in what aspects Theorem 2 differs from standard convex analysis? (maximization of an L-smooth concave function?).
Aren't there any other works that study convergence of guided diffusion models except Marion et al. [38]? There are several works on convergence of diffusion models, e.g., (De Bortoli, 2022). What prevents extending them to the guided case?

Valentin De Bortoli, "Convergence of denoising diffusion models under the manifold hypothesis," TMLR, 2022.

9.

Under the subspace assumption 1, your data covariance matrix is a DxD matrix of rank d<D.
Therefore it is not invertible. I suggest using the dagger symbol to avoid confusion between inverse and pseudoinverse. Provide more details in the proof to clarify that this issue is taken into account.
Note that there are some peculiar differences between inverse and pseudoinverse, e.g., generally, pinv(AB) is not equal to pinv(B)pinv(A).

10.

Regarding Section 5, you need to clearly state what are the differences of Alg 2 from many other works that fine-tune pretrained diffusion models (just google "adaptive diffusion" and "diffusion personalization").

11.

What is the motivation for convergence to the **global** optimum of the guidance objective if one pays in ignoring/omitting the prior information / regularization of the score model? (As seems to be promoted in Sections 5 and 6).
The tradeoff between reward/loss and regularization is trivial and clearly in most case there is some delicate balance between them that a user should look for.

---

> ### Author Rebuttal · Authors · 2024-08-07
>
> Thank you for appreciating our effort in providing theory for guidance. We first respond to your main concerns.
>
> >**Weakness 1** Assumptions made are strong. Any idea how can Theorem 1 be generalized to low dimensional manifold rather than low dimensional linear subspace?
>
> **A:** Given the challenge in developing theory for guided diffusion, adopting linear assuptions on data subspace and score funtion should be considered reasonable, more detailed discussion can be found in **Linear Assumption** part of [our rebuttal][re-link].
>
> Theorem 1 may have the potential to generalize to low-dimensional manifolds. For low-dimensional manifolds, the local geometry around any point can be approximated by its linear tangent space, and current Thm1 shows the guidance lies in this tangent space. By controlling the strength of guidance, the guidance vector on the tangent space will remain close to the manifold.
>
> > **Weakness 1 (Cont.)** In what aspects Thm 2 differ from standard convex analysis? There are  works on convergence of diffusion models, e.g., (De Bortoli, 2022). What prevents extending them to the guided case?
>
> **A:** **Diffusion theory cannot be simply extended to guided diffusion.** The addition of a guidance term disrupts the dynamics of the reverse SDE, thus the output distribution cannot be characterized. Previous theoretical work, such as [17](De Bortoli, 2022), heavily relies on the fact that distribution errors stem from approximation errors in the score function and the initial distribution of SDE. However, the training-free approach doesn't provide an approximation of $\nabla_{x_t} \log p_t(y|x_t)$ but instead introduces a plugin, implementation-convenient gradient guidance term. This discrepancy creates a barrier to directly extending existing theories to guided diffusion. [38](Marion et al, 2024) is the only work we know involving the theory of guided diffusion dynamics, and it also requires the linear score assumption. Therefore, this assumption should be considered reasonable. **Our key insight is characterizing the output sample distribution of guided diffusion as taking a proximal gradient step.** We derive the reverse process as a proximal step. This established relationship forms the premise for optimization and distinguishes our analysis from conventional convex optimization.
>
> >**Weakness 2:** A certain weakness of Theorems 2 and 3 is that they do not say anything about the variance.
>
> **A:** This is a misunderstanding, you might have missed the formula we provided for variance in line 235, which points to Eqn 25 in Appx E. This formula demonstrates that the variance of the output distribution is smaller than that of the training data (empirical covariance). Similarly, for Theorem 3, we present the variance in Eqn 31 of Appx E.3.
>
> >**Weakness 3:** There are no new algorithmic ideas in the paper, as the method is common, and also there exist works on fine-tuned/adaptive diffusion models.
>
> **A:**  Algorithmically, we introduce an iterative optimization algorithm that apply the prompted gradient guidance to the local linearization of objective function(Alg 1). This approach is new within the optimization context. In addition, our main focus is more on theoretical understaning for guided diffusion than proposing new method.
>
> Response to your other questions:
>
> >**Q1:** The statement in line 48 on "naive gradient guidance" is not clear. The explanation in Sec 3.2 that naive gradient doesn't work is not convincing because it ignores that fact that the step-size decreases and that the noise injection can mitigate error propagation.
>
> **A1:** We appreciate the reviewer's suggestion, we added a clear definition for "Naive gradient guidance" around line 48. To give a more convincing explanation for the failure of naive gradient-based guidance, we add a new lemma showing naive gradient suffers at least a constant error:
>
> **Lemma(Failure of naive guidance)** For naive guidance $\texttt{G}(X_t^{\leftarrow}, t) = b(t) \nabla f(X_t^{\leftarrow})$, suppose $b(t) >b_0>0$ for $t >t_0.$ For data in subspace under Assumption 1 and reward $f(x)=g^\top x$, $g \perp Span(A)$ with $h(t)=1-\exp(-\sqrt{t})$, then the orthogonal component of the generated sample is consistently large:
> $$
>     \mathbb{E} [X_{T,\perp}^{\leftarrow}] = C g, \quad C > \exp{\left(-5/2\right)}b_0.
> $$
> >**Proof.** Under Assumption 1, the score can be decomposed to terms parallel and orthogonal to Span(A) (Prop 2, Appx D.3) Applying naive guidance, we examine the orthogonal reverse process:
> $$
> \mathrm{d} X_{t, \perp}^{\leftarrow} =\left[\frac{1}{2}-\frac{1}{h(T-t)}\right] X_{t, \perp}^{\leftarrow}\mathrm{d} t + b(t)g \mathrm{d} t+\left(I_D-A A^\top\right) \mathrm{d} \overline{W}_t.
> $$
>
> >Solving this SDE, we get the expectation of the final state following $\mathbb{E}[X_{T, \perp}^{\leftarrow}] = \int_0^T \exp \left(- \int_0^{t}h^{-1}(s)\mathrm{d}s \right) e^{t/2}b(T-t) g \mathrm{d}t$. For $h(t) = 1 - \exp(-\sqrt{t})$, we have the coefficient of direction $g$ is larger than $\int_0^T \exp(-t/2-2\sqrt{t})b(T-t)\mathrm{d}t > \int_{0}^1 \exp(-5/2)b_0 \mathrm{d}t >0$ where we can assume $T>1$. Thus, $\mathbb{E}[ X_{T, \perp}^{\leftarrow}] \neq 0$. This means the generated sample is going out of the subspace, i.e., naive gradient guidance will violate the latent structure.
>
>
>
>
>
> [re-link]: https://openreview.net/forum?id=X1QeUYBXke&noteId=WsELdf2puF
> [pdf-link]: https://openreview.net/attachment?id=WsELdf2puF&name=pdf

---

> ### Author Response · Authors · 2024-08-07
>
> >**Q2:** The tight connection between gradient-guided diffusion models and proximal optimization algorithms is discussed in detail in Garber and Tirer (2024). This relevant literature should be mentioned.
>
> **A2:** This work is certainly related and has been included in our revised literature review (Sec 4 of [pdf][pdf-link]). However the connection between the proximal-gradient and gradient-based guidance discussed there is different from the connection we made. The connection in their paper is an optimization trick used to propose a two-step alternating algorithm for guided generation, which is classic in the research line of PnP (Plug in and Play). In contrast, our connection is an intuitive understanding distilled from our optimization theorem: pretrained model has an effect of regularization, guidance serves to optimize the objective.
>
> > **Q3:** Is the algorithm slow?
>
> **A3:** There might be misunderstanding on the reported "76 min" running time. We newly added a computation efficiency analysis with detailed running time records, please refer to **Table 1** and **computational efficiency** section in [our rebuttal][re-link] for more details.
>
> >**Q4:** What's the difference between Alg.2 and other fine-tuning diffusion models works?
>
> **A4:** Alg.2 simultaneously fine-tunes the pretrained model and adds a guidance term during inference. In contrast, existing works on fine-tuning diffusion models usually involves only fine-tuning the model weights.
> >**Q5:** What is the motivation for considering convergence to the global optimum of the objective if one pays in ignoring the prior information of the score model?
>
> **A5:** In our optimization framework, our goal is to find the global optimum **within the subspace**, which is also prior information or the underlying constraint. Thm 3 proves the exact convergence to the global optimum within the subspace. For example, in protein design, the generated samples should closely mimic natural proteins and comply with biological principles. Failure to do so can result in unstable structures, significantly degrading performance metrics. We agree that to which extent we balance reward optimization with regularization depends on the specific use case.
>
> >**Q6:**  Why do you state without loss of generality that $q=1$ for the noise schedule in line 85?
>
>
> **A6:** Our theoretical work actually holds for all general values of $q$, not just $q=1$. We appreciate your observation and we've clarified this point in our revision.
>
>
> >**Q7:** Why are Eqn 2 and Eqn 6 in [5] Song et al. (2020) inconsistent?
>
>
> **A7:** In our Eqn 2, $X_0$ represents the standard Gaussian noise distribution, whereas in Song et al.'s Eqn 6, $X_T$ represents the noise distribution. This difference in notation leads to a discrepancy in the $\mathrm{d}t$ term, specifically in its sign.
>
>
>
> >**Q8:** Suggests using the dagger symbol to distinguish between inverse and pseudoinverse.
>
> **A8:** Thank you for the suggestion. We will add more clarification in the revision. Regarding the data covariance matrix, it has the form $A\Sigma_u A^T$, where $\Sigma_u$ is full rank and the columns of $A$ are orthonormal, thus the pseudoinverse is $A\Sigma_u^{-1} A^T$.
>
> [re-link]: https://openreview.net/forum?id=X1QeUYBXke&noteId=WsELdf2puF
> [pdf-link]: https://openreview.net/attachment?id=WsELdf2puF&name=pdf

---

> > ### Comment · Reviewer_2yVt · 2024-08-11
> >
> > I thank the authors for their response, which addresses most of my concerns.
> > As mentioned above, I still believe that the claims on the algorithmic novelty should be toned done (e.g., the core guidance idea is common), and that already in the contribution list the strong assumption on the linear score model (Eq. 12) should be stated.
> > I decided to increase the rating, provided that the next version will include these changes, as well as discussion on the connection of gradient-guided diffusion schemes and proximal optimization methods that is stated in exiting work (where "pretrained model has an effect of regularization" as well).

---

> > > ### Author Response · Authors · 2024-08-11
> > > **Author Reply to Reviewer 2yVt**
> > >
> > > > I thank the authors for their response, which addresses most of my concerns.
> > >
> > > We are glad that our response addressed most of your concern and you increased the rating. Thank you again for raising some insightful questions, which help us improve the paper. With that being said, we will include in the revised text version: a discussion on similar guidance designs in existing literature, mentioning assumptions made in contribution list, as well as a discussion on the previously made connection between gradient-guided diffusion and proximal optimization, and its differece to the connection drawn in our paper.

---

### Official Review · Reviewer_8pJj · 2024-07-14

**Soundness:** 3
**Presentation:** 4
**Contribution:** 3
**Rating:** 5
**Confidence:** 3

**Summary:**

This paper rethinks the gradient-based guidance methods through the optimization perspective.  Similar to the manifold assumption, the authors first rely on the assumption that the observed data is from the lower dimensional space. Then, they claim that the naive gradient guidance does not maintain the data subspace, and show how to preserve the data subspace and incorporate guidance by using the look-ahead formulation. For the proof, they rely on the Gaussian linear model. From such motivations from theoretical results, they propose two versions of gradient guidance methods: 1) Only update the gradient guidance components and 2) Simultaneously finetuning the score model. These formulations enable the convergence to the data subspace, unlike naive guidance formulation. Experimental results are conducted on toy examples and image generation settings.

**Strengths:**

- Strong theories are presented in this paper. Rethinking the guidance formulation as the optimization procedure, that aims to optimize 1) guidance and 2) convergence of the data subspace, is interesting.
- The paper is well-written and easy to follow.

**Weaknesses:**

- While the proposed algorithm demonstrates considerable theoretical value, its practical implementation appears to be quite slow. This could be perceived as a weakness of the paper.
- The paper discusses Universal Guidance, yet it omits mention of similar training-free [B, C, D, E, F, G] and training-required guidance [A] mechanisms for utilizing off-the-shelf models. Including a discussion on these mechanisms would enhance the comprehensiveness of the paper. Additionally, it would be beneficial to address the practical and theoretical advantages compared to these mechanisms. As it stands, the paper does not sufficiently clarify these comparative advantages, which leaves some ambiguity about the relative strengths compared to these methods.

**Questions:**

- The paper relies heavily on the linear assumption. How reasonable is this assumption in the context of the proposed algorithm? Aren't guidance classifiers and score functions typically more complex than linear functions?

## **Reference**
- [A] Towards Practical Plug-and-Play Diffusion Models, CVPR 2023.

- [B] ELUCIDATING THE DESIGN SPACE OF CLASSIFIER GUIDED DIFFUSION GENERATION, ICLR 2024

- [C] ADJOINTDPM: ADJOINT SENSITIVITY METHOD FOR GRADIENT BACKPROPAGATION OF DIFFUSION PROBABILISTIC MODELS, ICLR 2024.

- [D] Towards Accurate Guided Diffusion Sampling through Symplectic Adjoint Method, Arxiv 2023.

- [E] Loss-guided Diffusion Models for Plug-and-Play Controllable Generation, ICML 2023

- [F] Manifold preserving guided diffusion, Arxiv 2023

- [G] Freedom: Training-free energy guided conditional diffusion model, ICCV 2023.

**Limitations:**

Linear approximation of the score function in their proof is mentioned as the limitation.

---

> ### Author Rebuttal · Authors · 2024-08-07
>
> Thank you for acknowledging our theoretical value and other comments! We first respond to your main concerns.
>
> >**Weakness 1:** While the proposed algorithm demonstrates considerable theoretical value, its practical implementation appears to be quite slow.
>
> **A1:** During rebuttal, we add a new running time analysis for our experiments, please refer to **Table 1** and the **computational efficiency** section in [our rebuttal][re-link] for more details.
>
>
> >**Weakness 2:** The paper discusses Universal Guidance, yet it omits mention of other similar training-free and training-required guidance mechanisms. Additionally, it would be beneficial to address the practical and theoretical advantages compared to these mechanisms.
>
>
> **A2:** We appreciate your mention of these related works that were omitted, and we've already included them in revision (Sec 4 in [pdf][pdf-link]) and discussed our practical and theoretical advantages compared to them. Here is a summary of the comparison.
>
> - First of all, to adapt a pre-trained diffusion model for guided generation, there are training-required methods [A] and training-free methods, which can be further classified into gradient guidance methods [E,F,G,B] and latent optimization methods [C,D]. Our method falls into gradient guidance methods.
>
> - Generally, classifier guidance [A] requires training/fine-tuning the classifier on noisy input even based on an off-the-shelf classifier, introducing extra training cost than training-free methods. Latent optimization methods are also training free, but are memory-intensive due to the need to store all intermediate gradients when utilizing the chain rule.
>
> - Within gradient-based methods, our work focuses on the theoretical aspect, providing understanding of the guidance method in [E,F,G,B]. Additionally, in our work, we are the first to iteratively apply the gradient guidance as a module for solving optimization problems, and provide theoretical guarantees.
>
> Response to your other questions:
>
> >**Question 1:** How reasonable is the assumption? Aren't guidance classifiers and score functions typically more complex than linear functions?
>
> A1: Please note that we do not assume the guidance classifier to be linear in our theorems. The form of our guidance is motivated by Gaussian data + linear classifier but its guarantees do not rely on the classifier to be a linear model. Thm 1 also holds for non-linear score, i.e., **arbitrary** distribution.
>
> We do assume linear score network in Thm 2 and 3, though being restrictive in practice, it is a reasonable assumption for establishing the first optimization theorem for guidance and yielding mathematical insights. The same assumption is also adopted by [38] (Marion et al, 2024), which is the only work we know involving the theory of guided diffusion dynamics.
> In addition, the effectiveness of our method is beyond linear assumption, justified by our experiments and new simulations (Sec 1 in [pdf][pdf-link] ) added during the rebuttal. More detailed discussion please refer to **Linear Assumption** section in [our rebuttal][re-link].
>
> [re-link]: https://openreview.net/forum?id=X1QeUYBXke&noteId=WsELdf2puF
> [pdf-link]: https://openreview.net/attachment?id=WsELdf2puF&name=pdf

---

> > ### Comment · Reviewer_8pJj · 2024-08-12
> >
> > I sincerely appreciate your thorough response. I carefully read all the reviewer's comments and discussions, and my concerns are addressed.

---

> ### Author Response · Authors · 2024-08-13
>
> Dear Reviewer 8pJj,
>
> We sincerely appreciate your insightful review and are glad we have successfully addressed your concerns. If you feel that all your questions have been satisfactorily answered, we would be grateful if you would consider raising the rating.
>
> Thanks again for your time and efforts in this work!
>
> Best regards,
>
> Authors,

---

### Author Rebuttal · Authors · 2024-08-07

We appreciate all reviewers for their valuable feedback!

### **New Theory, Experiments, and Running-Time Analysis**

**Theory for the failure of Naive gradient:** We construct a rigorous counterexample Sec 3 in [pdf][pdf-link] showing that the generated samples will suffer at least a constant error if the gradient of reward is directly applied as guidance.

**Experiment with data from nonlinear manifold:** We test our guidance on data from nonlinear manifold Sec 1 in [pdf][pdf-link] and plot the off-support deviation and reward curve. The curve shows that our guidance better preserves the data manifold than naive gradient guidance.

**Break-down analysis for running time:** We report and analyze the running time of our algorithms:

| | Total runtime (iterations)  | Per iteration | No guidance |
|------------|:-----------:|:-----------:|:-----------:|
| Simulation | **3.8 min** (50 iter),  76 min (1000 iter) | 4.6 s | 2.6 s |
| Image | **1.3 min**  (5 iter) |  15.8 s | 4.9 s |

**Table 1: Runtime Efficiency of Algorithm 1.** **Bold** refers to the total time to converge. No guidance refers to the time for one-time inference of the pre-trained model.

### **Expanded literature review**
We appreciate reviewers pointing us to several papers that are not directly under the topic of "diffusion model for optimization" but yet highly related. We include those in our revised literature review section in [pdf][pdf-link]. Gradient-based guidance and direct latent optimization are two main routes for adapting a pre-trained diffusion model to some reward, in a training-free way.  Our method falls into the class of gradient-based guidance, so we recap the comparison to other methods in this class here:

Gradient-based Guidance [A,B,C,D,E] utilize gradient as guidance. [A,B,D] solves inverse problem image and [C, E] solves guided/conditional image generation. [A,C] both propose similar guidance: on the predicted clean data $x_0$ with respect to $x_t$. Differently, our paper provides the first rigorous theoretical study of this gradient-based guidance from an optimization perspective, add prepose the first iterative algorithm that enjoys provable convergence guarantee.


### **Clarification on Our Contribution and Scope**

We want to highlight that our main contribution is to **provide the first optimization guarantee for adapting a pre-trained diffusion model to task-specific needs via gradient guidance**. Specifically: 1) We demonstrated the gradient guidance preserves learnt data structure (Thm 1); 2) We established optimization convergence theory for the gradient-guided diffusion model and its iterative fine-tuned variant (Thm2, 3; Fig 1).

### **Discussion on Linear Assumptions**
- **Linear case is a fundamental setting for theory and is highly nontrivial for theory on guided diffusion.** Extending diffusion theory to guided diffusion is highly nontrivial. The guidance term disrupts the reverse SDE dynamics which previous works in diffusion theory, such as [17] (De Bortoli, 2022), heavily rely on, making it impossible to characterize the output distribution if without further assumptions. [38] (Marion et al, 2024) is the only work we know involving the theory of guided diffusion dynamics, and it also requires the linear score assumption. Therefore, this assumption should be considered reasonable.

- **Practical usage and satisfying properties of our guidance and algorithms extend beyond linear assumption.**
    - **For assumption on linear score network:** When deploying, guidance *Gloss* is easily constructed for **any** pre-trained diffusion network, not limited to linear networks. Satisfying optimization performance was validated by experiments (Sec 6.1) with a 15M-parameter U-Net score network.
    - **For linear assumption on data manifold:** Theoretically, *Gloss* always preserves the data subspace for **arbitrary** latent distribution, not limited to Gaussians (Lem 1). For nonlinear data manifolds, our experiment (Sec 6.2) generated highly human-interpretable images, suggesting the manifold preserving property of *Gloss* stays for highly nonlinear manifolds. The new simulation in Sec 1 of [pdf][pdf-link] on sphere subspace also verified this property extending beyond linear assumption.

### **Clarification on Computation Efficiency**
Some reviewer misinterpret our discussions on computation efficiency that
> a single run of the backward sampling process (Module 1) takes 4.6s, and Alg. 1 takes 76min overall.

Here "76 min" is the total time for running to 1000 optimization rounds (one round corresponds to one complete reverse process with diffusion model), averaging 4.6 seconds per round in our simulations. Most of our simulations converge within 50 rounds, thus runs only 3.8 min to get satisfying results.

For Stable Diffusion v1.5, our approach needs 15.8 seconds per round. Typically, it achieves satisfactory results in fewer than 5 rounds (Figure 6), totaling 1.3 min. Reads are summarized in the **Table 1**.

**Backpropagation gradient through denoiser has been adopted by multiple gradient guidance method, thus it is NOT a deal breaker for efficiency.** Gradient-based guidance methods like [A,B,C] all require similar gradient backpropagation over the score network, thus our approach is not more computationally expensive in this regard.

References:

[A] Diffusion posterior sampling for general noisy inverse problems, 2022

[B] Loss-guided diffusion models for plug-and-play controllable generation, 2023

[C] Freedom: Training-free energy-guided conditional diffusion model. ICCV, 2023.

[D] Manifold preserving guided diffusion. arXiv preprint arXiv:2311.16424, 2023

[E] Towards accurate guided diffusion sampling through symplectic adjoint method. arXiv preprint arXiv:2312.12030, 2023b

[pdf-link]: https://openreview.net/attachment?id=WsELdf2puF&name=pdf

---

### Decision · Program_Chairs · 2024-09-25

**Decision:**

Accept (poster)

**Comment:**

The paper provides new theoretical and algorithmic insights into gradient guidance of diffusion models, which is the practice of adding an offset vector to the pre-trained score function estimate in the reverse SDE of a diffusion model with the goal of controlling the output toward a user-specified objective. The contributions are primarily theoretical and deal with guidance based on the “look-ahead” loss for estimating x0 given xt. The theory connects this type of gradient guidance to optimizing a user-specified loss function over the data subspace under the assumption that data lies on a linear subspace. The main convergence results are developed under the additional assumption that the pre-trained score-function estimator is a linear function of xt.

Reviewers unanimously appreciated the theory, calling it “interesting and novel”, “solid”, “very impressive”, etc.; Reviewer 2yVt appreciated that it provided theoretical evidence favoring one type of guidance over another. The primary concerns were about: (1) strong assumptions underlying the theoretical results, (2) efficiency in practice, (3) the tone of some claims in the introduction being too strong given the assumptions and/or prior evidence. The efficiency concerns were adequately addressed during the rebuttal. The assumptions and their implications were also discussed and some clarifications made. The meta-reviewer notes that assumptions were all clearly stated in the methods section, and all reviewers found the theory interesting and relevant despite not being completely general. I believe the issue is the matching of the claims in the into to the evidence provided by the theorems. I urge the authors to follow the suggestion of Reviewer 2yVt to temper the optimism surrounding the convergence results in the introduction by stating clearly in the introduction the assumption of a linear score network. I also echo the advice to more clearly reference prior evidence surrounding the use of “look-ahead” vs “naive” gradient guidance. With that said, it is clear that even with slightly toned down claims the paper provides valuable theoretical understanding for a fast-moving area.